# INFLUENCE FUNCTIONS FOR SCALABLE DATA ATTRIBUTION IN DIFFUSION MODELS

**Bruno Mlodozeniec**[*,1,2]  **Runa Eschenhagen**[1]  **Juhan Bae**[3,4]
**Alexander Immer**[2,5]  **David Krueger**[6]  **Richard Turner**[1,7]

[1]Department of Engineering, University of Cambridge, UK
[2]Max Planck Institute for Intelligent Systems, Tübingen, Germany
[3]Department of Computer Science, University of Toronto, Canada
[4]Vector Institute, Toronto, Canada
[5]Department of Computer Science, ETH Zurich, Switzerland
[6]Mila – Quebec AI Institute, Montreal, Canada
[7]The Alan Turing Institute, London, UK

## ABSTRACT

Diffusion models have led to significant advancements in generative modelling. Yet their widespread adoption poses challenges regarding data attribution and interpretability. In this paper, we aim to help address such challenges in diffusion models by developing an *influence function* framework. Influence function-based data attribution methods approximate how a model's output would have changed if some training data were removed. In supervised learning, this is usually used for predicting how the loss on a particular example would change. For diffusion models, we focus on predicting the change in the probability of generating a particular example via several proxy measurements. We show how to formulate influence functions for such quantities and how previously proposed methods can be interpreted as particular design choices in our framework. To ensure scalability of the Hessian computations in influence functions, we systematically develop K-FAC approximations based on generalised Gauss-Newton matrices specifically tailored to diffusion models. We recast previously proposed methods as specific design choices in our framework, and show that our recommended method outperforms previous data attribution approaches on common evaluations, such as the Linear Data-modelling Score (LDS) or retraining without top influences, without the need for method-specific hyperparameter tuning.

## 1 INTRODUCTION

Generative modelling for continuous data modalities — like images, video, and audio — has advanced rapidly, propelled by improvements in diffusion-based approaches. Many companies now offer easy access to AI-generated bespoke image content. However, the use of these models for commercial purposes creates a need for understanding how the training data influences their outputs. In cases where the model's outputs are undesirable, it is useful to be able to identify, and possibly remove, the training data instances responsible for those outputs. Furthermore, as copyrighted works often make up a significant part of the training corpora of these models (Schuhmann et al., 2022), concerns about the extent to which individual copyright owners' works influence the generated samples arise. Some already characterise what these companies offer as "copyright infringement as a service" (Saveri & Butterick, 2023a), which has caused a flurry of high-profile lawsuits Saveri & Butterick (2023a;b). This motivates exploring tools for data attribution that might be able to quantify how each group of training data points influences the models' outputs. Influence functions (Koh & Liang, 2017b; Bae

---

[*]Correspondence to: bkm28@cam.ac.uk
Source code available at https://github.com/BrunoKM/diffusion-influence

et al., 2022) offer precisely such a tool. By approximating the answer to the question, "If the model was trained with some of the data excluded, what would its output be?", they can help finding data points most responsible for a low loss on an example, or a high probability of generating a particular example. However, they have yet to be scalably adapted to the general diffusion modelling setting.

Influence functions work by locally approximating how the loss landscape would change if some of the training data points were down-weighted in the training loss (illustrated in Figure 4). Consequently, this enables prediction for how the (local) optimum of the training loss would change, and how that change in the parameters would affect a measurement of interest (e.g., loss on a particular example). By extrapolating this prediction, one can estimate what would happen if the data points were fully removed from the training set. However, to locally approximate the shape of the loss landscape, influence functions require computing and inverting the *Hessian* of the training loss, which is computationally expensive. One common approximation of the training loss's Hessian is the generalised Gauss-Newton matrix (GGN, Schraudolph, 2002; Martens, 2020). The GGN has not been clearly formulated for the diffusion modelling objective before and cannot be uniquely determined based on its general definition. Moreover, to compute and store a GGN for large neural networks further approximations are necessary. We propose using Kronecker-Factored Approximate Curvature (K-FAC, Heskes, 2000; Martens & Grosse, 2015) and its variant eigenvalue-corrected K-FAC (George et al., 2018, EK-FAC) to approximate the GGN. It is not commonly known how to apply it to neural network architectures used in diffusion models; for example, Kwon et al. (2023) resort to alternative Hessian approximation methods because "[K-FAC] might not be applicable to general deep neural network models as it highly depends on the model architecture". However, based on recent work, it is indeed clear that it can be applied to architectures used in diffusion models (Grosse & Martens, 2016; Eschenhagen et al., 2023), which typically combine linear layers, convolutions, and attention (Ho et al., 2020).

In this work, we describe a scalable approach to influence function-based approximations for data attribution in diffusion models, using (E)K-FAC approximation of GGNs as Hessian approximations. We articulate a design space based on influence functions, unify previous methods for data attribution in diffusion models (Georgiev et al., 2023; Zheng et al., 2024) through our framework, and argue for the design choices that distinguish our method from previous ones. One important design choice is the GGN used as the Hessian approximation. We formulate different GGN matrices for the diffusion modelling objective and discuss their implicit assumptions. We empirically ablate variations of the GGN approximation and other design choices in our framework and show that our proposed method outperforms the existing data attribution methods for diffusion models as measured by common data attribution metrics like the Linear Datamodeling Score (Park et al., 2023) or retraining without top influences. Finally, we also discuss interesting empirical observations that challenge our current understanding of influence functions in the context of diffusion models.

## 2 BACKGROUND

This section introduces the general concepts of diffusion models, influence functions, and the GGN.

### 2.1 DIFFUSION MODELS

Diffusion models are a class of probabilistic generative models that fit a model $p_\theta(x)$ parameterised by parameters $\theta \in \mathbb{R}^{d_{\text{param}}}$ to approximate a training data distribution $q(x)$, with the primary aim being to sample new data $x \sim p_\theta(\cdot)$ (Sohl-Dickstein et al., 2015; Ho et al., 2020; Turner et al., 2024). This is usually done by augmenting the original data $x$ with $T$ fidelity levels as $x^{(0:T)} = [x^{(0)}, \ldots, x^{(T)}]$ with an augmentation distribution $q(x^{(0:T)})$ that satisfies the following criteria: **1)** the highest fidelity $x^{(0)}$ equals the original training data $q(x^{(0)}) = q(x)$, **2)** the lowest fidelity $x^{(T)}$ has a distribution that is easy to sample from, and **3)** predicting a lower fidelity level from the level directly above it is simple to model and learn. To achieve the above goals, $q$ is typically taken to be a first-order Gaussian auto-regressive (diffusion) process: $q(x^{(t)}|x^{(0:t-1)}) = \mathcal{N}(x^{(t)}|\lambda_t x^{(t-1)}, (1 - \lambda_t)^2 I)$, with hyperparameters $\lambda_t$ set so that the law of $x^{(T)}$ approximately matches a standard Gaussian distribution $\mathcal{N}(0, I)$. In that case, the reverse conditionals $q(x^{(t-1)}|x^{(t:T)}) = q(x^{(t-1)}|x^{(t)})$ are first-order Markov, and if the number of fidelity levels $T$ is high enough, they can be well approximated by a diagonal Gaussian, allowing them to be modelled with a parametric model with a

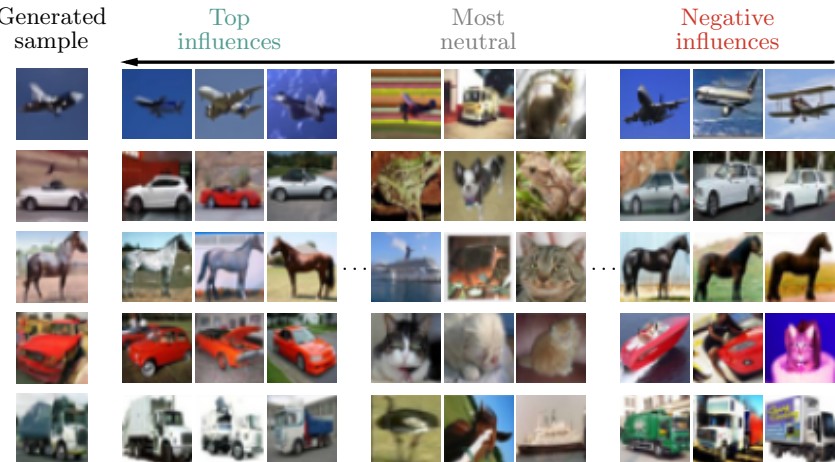

Figure 1: Most influential training data points as identified by K-FAC Influence Functions for samples generated by a denoising diffusion probabilistic model trained on `CIFAR-10`. The top influences are those whose omission from the training set is predicted to most increase the loss of the generated sample. Negative influences are those predicted to most decrease the loss, and the most neutral are those that should change the loss the least.

simple likelihood function, hence satisfying (3) (Turner et al., 2024). The marginals $q(x^{(t)}|x^{(0)}) = \mathcal{N}\left(x^{(t)}|\left(\prod_{t'=1}^{t}\lambda_{t'}\right)x^{(0)}, \left(1 - \prod_{t'=1}^{t}\lambda_{t'}^2\right)I\right)$ also have a simple Gaussian form, allowing for the augmented samples to be sampled as:

$$x^{(t)} = \prod_{t'=1}^{t}\lambda_t x^{(0)} + \left(1 - \prod_{t'=1}^{t}\lambda_{t'}^2\right)^{1/2}\epsilon^{(t)}, \qquad \text{with } \epsilon^{(t)} \sim \mathcal{N}(0, I). \qquad (1)$$

Diffusion models are trained to approximate the reverse conditionals $p_\theta(x^{(t-1)}|x^{(t)}) \approx q(x^{(t-1)}|x^{(t)})$ by maximising log-probabilities of samples $x^{(t-1)}$ conditioned on $x^{(t)}$, for all timesteps $t = 1, \ldots, T$. We can note that $q(x^{(t-1)}|x^{(t)}, x^{(0)})$ has a Gaussian distribution with mean given by:

$$\mu_{t-1|t,0}(x^{(t)}, \epsilon^{(t)}) = \frac{1}{\lambda_t}\left(x^{(t)} - \frac{1 - \lambda_t^2}{\left(1 - \prod_{t'=1}^{t}\lambda_{t'}^2\right)^{1/2}}\epsilon^{(t)}\right), \qquad \text{with } \epsilon^{(t)} \stackrel{\text{def}}{=} \frac{(x^{(t)} - \prod_{t'=1}^{t}\lambda_{t'}x^{(0)})}{(1 - \prod_{t'=1}^{t}\lambda_{t'}^2)^{1/2}}$$

as in Equation (1). In other words, the mean is a mixture of the sample $x^{(t)}$ and the noise $\epsilon^{(t)}$ that was applied to $x^{(0)}$ to produce it. Hence, we can choose to analogously parameterise $p_\theta(x^{(t-1)}|x^{(t)})$ as $\mathcal{N}\left(x^{(t-1)}|\mu_{t-1|t,0}\left(x^{(t)}, \epsilon_\theta^t(x^{(t)})\right), \sigma_t^2 I\right)$. That way, the model $\epsilon_\theta^{(t)}(x^{(t)})$ simply predicts the noise $\epsilon^{(t)}$ that was added to the data to produce $x^{(t)}$. The variances $\sigma_t^2$ are usually chosen as hyperparameters (Ho et al., 2020). With that parameterisation, the negative expected log-likelihood $\mathbb{E}_{q(x^{t-1}, x^{(t)}|x^{(0)})}\left[-\log p(x^{(t-1)}|x^{(t)})\right]$, up to scale and shift independent of $\theta$ or $x^{(0)}$, can be written as (Ho et al., 2020; Turner et al., 2024):[1]

$$\ell_t(\theta, x^{(0)}) = \mathbb{E}_{\epsilon^{(t)}, x^{(t)}}\left[\left\|\epsilon^{(t)} - \epsilon_\theta^t\left(x^{(t)}\right)\right\|^2\right] \qquad \begin{aligned}\epsilon^{(t)} &\sim \mathcal{N}(0, I) \\ x^{(t)} &= \prod_{t'=1}^{t}\lambda_t x^{(0)} + \left(1 - \prod_{t'=1}^{t}\lambda_{t'}^2\right)^{1/2}\epsilon^{(t)}\end{aligned} \qquad (2)$$

This leads to a training loss $\ell$ for the diffusion model $\epsilon_\theta^t(x^{(t)})$ that is a sum of per-diffusion timestep training losses:[2]

$$\ell(\theta, x) = \mathbb{E}_{\tilde{t}}\left[\ell_{\tilde{t}}(\theta, x)\right] \qquad \tilde{t} \sim \text{Uniform}([T]).$$

The parameters are then optimised to minimise the loss averaged over a training dataset $\mathcal{D} = \{x_n\}_{n=1}^N$:

$$\theta^\star(\mathcal{D}) = \arg\min_\theta \mathcal{L}_\mathcal{D}(\theta) \quad \mathcal{L}_\mathcal{D}(\theta) \stackrel{\text{def}}{=} \frac{1}{N}\sum_{n=1}^N \ell(\theta, x_n). \qquad (3)$$

---

[1] Note that the two random variables $x^{(t)}, \epsilon^{(t)}$ are deterministic functions of one-another.

[2] Equivalently, a weighted sum of per-timestep negative log-likelihoods $-\log p_\theta(x^{(t-1)}|x^{(t)})$.

Other interpretations of the above procedure exist in the literature (Song & Ermon, 2020; Song et al., 2021b;a; Kingma et al., 2023).

## 2.2 INFLUENCE FUNCTIONS

The aim of influence functions is to answer questions of the sort "how would my model behave were it trained on the training dataset with some datapoints removed". To do so, they approximate the change in the optimal model parameters in Equation (3) when some training examples $(x_j)_{j \in \mathcal{I}}$, $\mathcal{I} = \{i_1, \dots, i_M\} \subseteq [N]$, are removed from the dataset $\mathcal{D}$. To arrive at a tractable approximation, it is useful to consider a continuous relaxation of this question: how would the optimum change were the training examples $(x_j)_{j \in \mathcal{I}}$ down-weighted by $\varepsilon \in \mathbb{R}$ in the training loss:

$$r_{-\mathcal{I}}(\varepsilon) = \arg\min_{\theta} \frac{1}{N} \sum_{n=1}^{N} \ell(\theta, x_n) - \varepsilon \sum_{j \in \mathcal{I}} \ell(\theta, x_j) \qquad (4)$$

The function $r_{-\mathcal{I}} : \mathbb{R} \to \mathbb{R}^{d_{\text{param}}}$ (well-defined if the optimum is unique) is the *response function*. Setting $\varepsilon$ to $1/N$ recovers the minimum of the original objective in Equation (3) with examples $(x_{i_1}, \dots, x_{i_M})$ removed.

Under suitable assumptions (see Appendix A), by the Implicit Function Theorem (Krantz & Parks, 2003), the response function is continuous and differentiable at $\varepsilon = 0$. *Influence functions* can be defined as a linear approximation to the response function $r_{-\mathcal{I}}$ by a first-order Taylor expansion around $\varepsilon = 0$:

$$\begin{aligned} r_{-\mathcal{I}}(\varepsilon) = r_{-\mathcal{I}}(0) \quad &+ \left. \frac{dr_{-\mathcal{I}}(\varepsilon')}{d\varepsilon'} \right|_{\varepsilon'=0} \varepsilon &+ o(\varepsilon) \\ = \theta^{\star}(\mathcal{D}) \quad &+ \sum_{j \in \mathcal{I}} \left( \nabla_{\theta^{\star}}^2 \mathcal{L}_{\mathcal{D}}(\theta^{\star}) \right)^{-1} \nabla_{\theta^{\star}} \ell(\theta^{\star}, x_j) \varepsilon &+ o(\varepsilon), \end{aligned} \qquad (5)$$

as $\varepsilon \to 0$. See Appendix A for a formal derivation and conditions. The optimal parameters with examples $(x_i)_{i \in \mathcal{I}}$ removed can be approximated by setting $\varepsilon$ to $1/N$ and dropping the $o(\varepsilon)$ terms.

Usually, we are not directly interested in the change in parameters in response to removing some data, but rather the change in some *measurement* function $m(\theta^{\star}(\mathcal{D}), x')$ at a particular test input $x'$ (e.g., per-example test loss). We can further make a first-order Taylor approximation to $m(\cdot, x')$ at $\theta^{\star}(\mathcal{D})$ — $m(\theta, x') = m(\theta^{\star}, x') + \nabla_{\theta^{\star}}^{\mathsf{T}} m(\theta^{\star}, x')(\theta - \theta^{\star}) + o(\|\theta - \theta^{\star}\|_2)$ — and combine it with Equation (5) to get a simple linear estimate of the change in the measurement function:

$$m(r_{-\mathcal{I}}(\varepsilon), x') = m(\theta^{\star}, x') + \sum_{j \in \mathcal{I}} \nabla_{\theta^{\star}}^{\mathsf{T}} m(\theta^{\star}, x') \left( \nabla_{\theta^{\star}}^2 \mathcal{L}_{\mathcal{D}}(\theta^{\star}) \right)^{-1} \nabla_{\theta^{\star}} \ell(\theta^{\star}, x_j) \varepsilon + o(\varepsilon). \qquad (6)$$

## 2.3 GENERALISED GAUSS-NEWTON MATRIX

Computing the influence function approximation in Equation (5) requires inverting the Hessian $\nabla_{\theta}^2 \mathcal{L}_{\mathcal{D}}(\theta) \in \mathbb{R}^{d_{\text{param}} \times d_{\text{param}}}$. In the context of neural networks, the Hessian itself is generally computationally intractable and approximations are necessary. A common Hessian approximation is the generalised Gauss-Newton matrix (GGN). We will first introduce the GGN in an abstract setting of approximating the Hessian for a general training loss $\mathcal{L}(\theta) = \mathbb{E}_z[\rho(\theta, z)]$, to make it clear how different variants can be arrived at for diffusion models in the next section.

In general, if we have a function $\rho(\theta, z)$ of the form $h_z \circ f_z(\theta)$, with $h_z$ a convex function, the GGN for an expectation $\mathbb{E}_z[\rho(\theta, z)]$ is defined as

$$\text{GGN}(\theta) = \mathbb{E}_z \left[ \nabla_{\theta}^{\mathsf{T}} f_z(\theta) \left( \nabla_{f_z(\theta)}^2 h_z(f_z(\theta)) \right) \nabla_{\theta} f_z(\theta) \right],$$

where $\nabla_{\theta} f_z(\theta)$ is the Jacobian of $f_z$. Whenever $f_z$ is (locally) linear, the GGN is equal to the Hessian $\mathbb{E}_z[\nabla_{\theta}^2 \rho(\theta, z)]$. Therefore, we can consider the GGN as an approximation to the Hessian in which we "linearise" the function $f_z$. Note that any decomposition of $\rho(\theta, z)$ results in a valid GGN as long as $h_z$ is convex (Martens, 2020).[3] We give two examples below.

---

[3] $h_z$ is typically required to be convex to guarantee the resulting GGN is a positive semi-definite (PSD) matrix. A valid non-PSD approximation to the Hessian can be formed with a non-convex $h_z$ as well; all the arguments about the exactness of the GGN approximation for a linear $f_z$ would still apply. However, the PSD property helps with numerical stability of the matrix inversion, and guarantees that the GGN will be invertible if a small damping term is added to the diagonal.

**Option 1.** A typical choice would be for $f_z$ to be the neural network function on a training datapoint $z$, and for $h_z$ to be the loss function (e.g. $\ell_2$-loss), with the expectation $\mathbb{E}_z$ being taken over the empirical (training) data distribution; we call the GGN for this split $\text{GGN}^{\texttt{model}}$. The GGN with this split is exact for linear neural networks (or when the model has zero residuals on the training data) (Martens, 2020).

$$
\begin{aligned}
f_z &:= \text{mapping from parameters to model output} \\
h_z &:= \text{loss function (e.g. } \ell_2\text{-loss)}
\end{aligned}
\quad \rightarrow \text{GGN}^{\texttt{model}}_\theta(\theta) \tag{7}
$$

**Option 2.** Alternatively, a different GGN can be defined by using a trivial split of the loss $\rho(\theta, z)$ into the log map $h_z := -\log$ and the exponentiated negated loss $f_z := \exp(-\rho(\cdot, z))$, and again taking the expectation over the empirical data distribution. With this split, the resulting GGN is

$$
\begin{aligned}
f_z &:= \exp(-\rho(\cdot, z)) \\
h_z &:= -\log
\end{aligned}
\quad \rightarrow \text{GGN}^{\texttt{loss}}(\theta) = \mathbb{E}_z \left[ \nabla_\theta \rho(\theta, z) \nabla_\theta^\intercal \rho(\theta, z) \right]. \tag{8}
$$

This is also called the empirical Fisher (Kunstner et al., 2019). Note that $\text{GGN}^{\texttt{loss}}$ is only equal to the Hessian under the arguably more stringent condition that $\exp(-\rho(\cdot, z))$ — the composition of the model *and* the exponentiated negative loss function — is linear. This is in contrast to $\text{GGN}^{\texttt{model}}$, for which only the mapping from the parameters to the model output needs to be (locally) linear. Hence, we might prefer to use $\text{GGN}^{\texttt{model}}$ for Hessian approximation whenever we have a nonlinear loss, which is the case for diffusion models.

## 3 SCALABLE INFLUENCE FUNCTIONS FOR DIFFUSION MODELS

In this section, we discuss how we adapt influence functions to the diffusion modelling setting in a scalable manner. We also recast data attribution methods for diffusion models proposed in prior work (Georgiev et al., 2023; Zheng et al., 2024) as the result of particular design decisions in our framework, and argue for our own choices that distinguish our method from the previous ones.

### 3.1 APPROXIMATING THE HESSIAN

In diffusion models, we want to compute the Hessian of the loss of the form

$$
\mathcal{L}_\mathcal{D}(\theta) = \mathbb{E}_{x_n} \left[ \ell(\theta, x_n) \right] = \mathbb{E}_{x_n} \left[ \mathbb{E}_{\tilde{t}} \left[ \mathbb{E}_{x^{(\tilde{t})}, \epsilon^{(\tilde{t})}} \left[ \| \epsilon^{(\tilde{t})} - \epsilon_\theta^{\tilde{t}}(x^{(\tilde{t})}) \|^2 \right] \right] \right],
$$

where $\mathbb{E}_{x_n}[\cdot] = \left( \frac{1}{N} \sum_{n=1}^N \cdot \right)$ is the expectation over the empirical data distribution. [4] We will describe how to formulate different GGN approximations for this setting.

#### 3.1.1 GGN FOR DIFFUSION MODELS

**Option 1.** To arrive at a GGN approximation, as discussed in Section 2.3, we can partition the function $\theta \mapsto \| \epsilon^{(t)} - \epsilon_\theta^t(x^{(t)}) \|^2$ into the model output $\theta \mapsto \epsilon_\theta^t(x^{(t)})$ and the $\ell_2$-loss function $\| \epsilon^{(t)} - \cdot \|^2$. This results in the GGN:

$$
\begin{aligned}
f_z &:= \epsilon_\theta^{\tilde{t}}(x^{(\tilde{t})}) \\
h_z &:= \| \epsilon^{(\tilde{t})} - \cdot \|^2
\end{aligned}
\quad \rightarrow \text{GGN}^{\texttt{model}}_\mathcal{D}(\theta) = \mathbb{E}_{x_n} \left[ \mathbb{E}_{\tilde{t}} \left[ \mathbb{E}_{x^{(\tilde{t})}, \epsilon^{(\tilde{t})}} \left[ \nabla_\theta^\intercal \epsilon_\theta^{\tilde{t}} \left( x^{(\tilde{t})} \right) (2I) \nabla_\theta \epsilon_\theta^{\tilde{t}} \left( x^{(\tilde{t})} \right) \right] \right] \right], \tag{9}
$$

where $I$ is the identity matrix. This correspond to "linearising" the neural network $\epsilon_\theta^t$. For diffusion models, the dimensionality of the output of $\epsilon_\theta^{\tilde{t}}$ is typically very large (e.g. $32 \times 32 \times 3$ for CIFAR), so computing the Jacobians $\nabla_\theta \epsilon_\theta^t$ explicitly is still intractable. However, we can express $\text{GGN}^{\texttt{model}}_\mathcal{D}$ as

$$
\text{F}_\mathcal{D}(\theta) = \mathbb{E}_{x_n} \left[ \mathbb{E}_{\tilde{t}} \left[ \mathbb{E}_{x_n^{(\tilde{t})}} \left[ \mathbb{E}_{\epsilon_{\texttt{mod}}} \left[ g_n(\theta) g_n(\theta)^\intercal \right] \right] \right] \right], \qquad \epsilon_{\texttt{mod}} \sim \mathcal{N} \left( \epsilon_\theta^{\tilde{t}} \left( x_n^{(\tilde{t})} \right), I \right) \tag{10}
$$

where $g_n(\theta) = \nabla_\theta \| \epsilon_{\texttt{mod}} - \epsilon_\theta^{\tilde{t}}(x_n^{(\tilde{t})}) \|^2 \in \mathbb{R}^{d_{\texttt{param}}}$; see Appendix B for the derivation. This formulation lends itself to a Monte Carlo approximation, since we can now compute gradients using auxiliary

---

[4]Generally, $\mathbb{E}_{x_n}$ might also subsume the expectation over data augmentations applied to the training data points (see Appendix J.9 for details on how this is handled).

targets $\epsilon_{\texttt{mod}}$ sampled from the model's output distribution, as shown in Equation (10). $\mathrm{F}_{\mathcal{D}}$ can be interpreted as a kind of Fisher information matrix (Amari, 1998; Martens, 2020), but it is not the Fisher for the marginal model distribution $p_\theta(x)$.

**Option 2.** Analogously to Equation (8), we can also consider the trivial decomposition of $\ell(\cdot, x)$ into the $\log$ and the exponentiated loss, effectively "linearising" $\exp(-\ell(\cdot, x))$. The resulting GGN is:

$$
\begin{aligned}
f_z &:= \exp(-\ell(\cdot, x_n)) \\
h_z &:= -\log
\end{aligned}
\quad \rightarrow \mathrm{GGN}_{\mathcal{D}}^{\texttt{loss}}(\theta) = \mathbb{E}_{x_n}\left[\nabla_\theta \ell(\theta, x_n) \nabla_\theta^\intercal \ell(\theta, x_n)\right],
\tag{11}
$$

where $\ell(\theta, x)$ is the diffusion training loss defined in Equation (2). This Hessian approximation $\mathrm{GGN}_{\mathcal{D}}^{\texttt{loss}}$ turns out to be equivalent to the ones considered in the previous works on data attribution for diffusion models (Georgiev et al., 2023; Zheng et al., 2024; Kwon et al., 2023). In contrast, in this work, we opt for $\mathrm{GGN}_{\mathcal{D}}^{\texttt{model}}$ in Equation (9), or equivalently $\mathrm{F}_{\mathcal{D}}$, since it is arguably a better-motivated approximation of the Hessian than $\mathrm{GGN}_{\mathcal{D}}^{\texttt{loss}}$ (c.f. Section 2.3).

In Zheng et al. (2024), the authors explored substituting different (theoretically incorrect) training loss functions into the influence function approximation. In particular, they found that replacing the loss $\|\epsilon^{(t)} - \epsilon_\theta^t(x^{(t)})\|^2$ with the square norm loss $\|\epsilon_\theta^t(x^{(t)})\|^2$ (effectively replacing the "targets" $\epsilon^{(t)}$ with 0) gave the best results. Note that the targets $\epsilon^{(t)}$ do not appear in the expression for $\mathrm{GGN}_{\mathcal{D}}^{\texttt{model}}$ in Equation (9).[5] Hence, in our method substituting different targets would not affect the Hessian approximation. In Zheng et al. (2024), replacing the targets only makes a difference to the Hessian approximation because they use $\mathrm{GGN}_{\mathcal{D}}^{\texttt{loss}}$ (an empirical Fisher) to approximate the Hessian.

### 3.1.2 (E)K-FAC FOR DIFFUSION MODELS

While $\mathrm{F}_{\mathcal{D}}(\theta)$ and $\mathrm{GGN}_{\mathcal{D}}^{\texttt{loss}}$ do not require computing full Jacobians or the Hessian of the neural network model, they involve taking outer products of gradients in $\mathbb{R}^{d_{\texttt{param}}}$, which is still intractable. Kronecker-Factored Approximate Curvature (Heskes, 2000; Martens & Grosse, 2015, K-FAC) is a common scalable approximation of the GGN to overcome this problem. It approximates the GGN with a block-diagonal matrix, where each block corresponds to one neural network layer and consists of a Kronecker product of two matrices. Due to convenient properties of the Kronecker product, this makes the inversion and multiplication with vectors needed in Equation (6) efficient enough to scale to large networks. K-FAC is defined for linear layers, including linear layers with weight sharing like convolutions (Grosse & Martens, 2016). This covers most layer types in the architectures typically used for diffusion models (linear, convolutions, attention). When weight sharing is used, there are two variants – K-FAC-expand and K-FAC-reduce (Eschenhagen et al., 2023); see Appendix C.1 for an overview. For the parameters $\theta_l$ of layer $l$, the GGN $\mathrm{F}_{\mathcal{D}}$ in Equation (10) is approximated by

$$
\mathrm{F}_{\mathcal{D}}(\theta_l) \approx \frac{1}{N^2} \sum_{n=1}^{N} \mathbb{E}_{\tilde{t}}\left[\mathbb{E}_{x_n^{(\tilde{t})}, \epsilon^{(\tilde{t})}}\left[a_n^{(l)} a_n^{(l)\intercal}\right]\right] \otimes \sum_{n=1}^{N} \mathbb{E}_{\tilde{t}}\left[\mathbb{E}_{x_n^{(\tilde{t})},, \epsilon^{(\tilde{t})}, \epsilon_{\texttt{mod}}^{(\tilde{t})}}\left[b_n^{(l)} b_n^{(l)\intercal}\right]\right],
\tag{12}
$$

with $a_n^{(l)} \in \mathbb{R}^{d_{\texttt{in}}^l}$ being the inputs to the $l$th layer for data point $x_n^{(\tilde{t})}$ and $b_n^{(l)} \in \mathbb{R}^{d_{\texttt{out}}^l}$ being the gradient of the $\ell_2$-loss w.r.t. the output of the $l$th layer, and $\otimes$ denoting the Kronecker product.[6] The approximation trivially becomes an equality for a single data point and also for deep linear networks with $\ell_2$-loss (Bernacchia et al., 2018; Eschenhagen et al., 2023).

For our recommended method, we choose to approximate the Hessian with a K-FAC approximation of $\mathrm{F}_{\mathcal{D}}$, akin to Grosse et al. (2023). We approximate the expectations in Equation (12) with Monte Carlo samples and use K-FAC-expand whenever weight sharing is used; in the case of convolutional layers this corresponds to Grosse & Martens (2016). See C.2 for the full derivation of K-FAC for diffusion models that also considers weight sharing. Additionally, we choose to use eigenvalue-corrected K-FAC (George et al., 2018, EK-FAC) in our experiments — as suggested by Grosse et al. (2023) — which improves performance notably and can be directly applied on top of our K-FAC approximation. Lastly, to ensure the Hessian approximation is well-conditioned and invertible, we follow standard practice and add a damping term consisting of a small scalar damping factor times the identity matrix. We ablate all of these design choices in Appendices C.3 and G (Figures 5, 7 and 9).

---

[5]This is because the Hessian of an $\ell_2$-loss w.r.t. the model output is a multiple of the identity matrix.

[6]For the sake of a simpler presentation this does not take potential weight sharing into account.

## 3.2 GRADIENT COMPRESSION AND QUERY BATCHING

In practice, we recommend computing influence function estimates in Equation ([6](#)) by first computing and storing the approximate Hessian inverse, and then iteratively computing the preconditioned inner products $\nabla_{\theta^\star}^\intercal m(\theta^\star, x) \left(\nabla_{\theta^\star}^2 \mathcal{L}_\mathcal{D}(\theta^\star)\right)^{-1} \nabla_{\theta^\star} \ell(\theta^\star, x_j)$ for different training datapoints $x_j$. Following Grosse et al. (2023), we use query batching to avoid recomputing the gradients $\nabla_{\theta^\star} \ell(\theta^\star, x_j)$ when attributing multiple samples $x$. We also use gradient compression; we found that compression by quantisation works much better for diffusion models compared to the SVD-based compression used by Grosse et al. (2023) (see Appendix [F](#)), likely due to the fact that gradients $\nabla_\theta \ell(\theta, x_n)$ are not low-rank in this setting.

## 3.3 WHAT TO MEASURE

For diffusion models, arguably the most natural question to ask might be, for a given sample $x$ generated from the model, how did the training samples influence the probability of generating a sample $x$? For example, in the context of copyright infringement, we might want to ask if removing certain copyrighted works would substantially reduce the probability of generating $x$. With influence functions, these questions could be interpreted as setting the measurement function $m(\theta, x)$ to be the (marginal) log-probability of generating $x$ from the diffusion model: $\log p_\theta(x)$.

Computing the marginal log-probability introduces some challenges. Diffusion models have originally been designed with the goal of tractable sampling, and not log-likelihood evaluation. Ho et al. (2020); Sohl-Dickstein et al. (2015) only introduce a lower-bound on the marginal log-probability. Song et al. (2021b) show that exact log-likelihood evaluation is possible, but it only makes sense in settings where the training data distribution has a density (e.g. uniformly dequantised data), and it only corresponds to the marginal log-likelihood of the model when sampling deterministically (Song et al., 2021a).[7] Also, taking gradients of that measurement, as required for influence functions, is non-trivial. Hence, in most cases, we might need a proxy measurement for the marginal probability. We consider a couple of proxies in this work:

1. **Loss.** Approximate $\log p_\theta(x)$ with the diffusion loss $\ell(\theta, x)$ in Equation ([2](#)) on that particular example. This corresponds to the ELBO with reweighted per-timestep loss terms (see Figure [20](#)).
2. **Probability of sampling trajectory.** If the entire sampling trajectory $x^{(0:T)}$ that generated sample $x$ is available, consider the probability of that trajectory $p_\theta(x^{(0:T)}) = p(x^T) \prod_{t=1}^T p_\theta(x^{(t-1)}|x^{(t)})$.
3. **ELBO.** Approximate $\log p_\theta(x)$ with an Evidence Lower-Bound (Ho et al., 2020, eq. (5)).

## 4 EXPERIMENTS

**Evaluating Data Attribution.** To evaluate the proposed data attribution methods, we primarily focus on two metrics: *Linear Data Modelling Score* (LDS) and *retraining without top influences*. These metrics are described below. In all experiments, we look at measurements on samples generated by the model trained on $\mathcal{D}$.[8] We primarily focus on Denoising Diffusion Probabilistic Models (DDPM) (Ho et al., 2020) throughout. Runtimes are reported in Appendix [E](#).

LDS measures how well a given attribution method predicts the relative change in a measurement as the model is retrained on (random) subsets of the training data. For an attribution method $a(\mathcal{D}, \mathcal{D}', x)$ that approximates how a measurement $m(\theta^\star(\mathcal{D}), x)$ would change if a model was trained on an altered dataset $\mathcal{D}'$, LDS measures the Spearman rank correlation between the predicted changes in output $a(\mathcal{D}, \tilde{\mathcal{D}}_1, x), \ldots, a(\mathcal{D}, \tilde{\mathcal{D}}_M, x)$ and the actual changes in output $m(\theta^\star(\tilde{\mathcal{D}}_1), x), \ldots, m(\theta^\star(\tilde{\mathcal{D}}_M), x)$ after retraining on $M$ independently subsampled versions $\tilde{\mathcal{D}}_i$ of the original dataset $\mathcal{D}$, each containing $50\%$ of the points sampled without replacement. However, training on a fixed dataset can produce different models with functionally different behaviour depending on the random seed used for the initialisation and data order during stochastic optimisation. Hence, for any given dataset $\mathcal{D}'$, different

---

[7]Unless the trained model satisfies very specific "consistency" constraints (Song et al., 2021b, Theorem 2).

[8]Higher LDS values can be obtained when looking at validation examples (Zheng et al., 2024), but diffusion models are used primarily for sampling, so attributing generated samples is of primary practical interest.

measurements could be obtained depending on the random seed used. To mitigate the issue, Park et al. (2023) suggest using an ensemble average measurement after retraining as the "oracle" target:

$$\text{LDS} = \text{spearman}\left[\left(a(\mathcal{D}, \tilde{\mathcal{D}}_i, x)\right)_{i=1}^{M}; \left(\frac{1}{K}\sum_{k=1}^{K} m(\tilde{\theta}_k^{\star}(\tilde{\mathcal{D}}_i), x)\right)_{i=1}^{M}\right], \tag{13}$$

where $\tilde{\theta}_k^{\star}(\mathcal{D}') \in \mathbb{R}^{d_{\texttt{param}}}$ are the parameters resulting from training on $\mathcal{D}'$ with a particular seed $k$.

Retraining without top influences, on the other hand, evaluates the ability of the data attribution method to surface the most influential data points – namely, those that would most negatively affect the measurement $m(\theta^{\star}(\mathcal{D}'), x)$ under retraining from scratch on a dataset $\mathcal{D}'$ with these data points removed. For each method, we remove a fixed percentage of the most influential datapoints from $\mathcal{D}$ to create the new dataset $\mathcal{D}'$, and report the change in the measurement $m(\theta^{\star}(\mathcal{D}'), x)$ relative to $m(\theta^{\star}(\mathcal{D}), x)$ (measurement by the model trained on the full dataset $\mathcal{D}$).

**Methods.** We compare influence functions with EK-FAC and $\text{GGN}_{\mathcal{D}}^{\texttt{model}}$ (MC-Fisher; Equation (10)) as the Hessian approximation (termed **K-FAC Influence**) to TRAK as formulated for diffusion models in Georgiev et al. (2023); Zheng et al. (2024). In our framework, their method can be tersely described as using $\text{GGN}_{\mathcal{D}}^{\texttt{loss}}$ (Empirical Fisher) in Equation (11) as a Hessian approximation instead of $\text{GGN}_{\mathcal{D}}^{\texttt{model}}$ (MC-Fisher) in Equation (10), and computing the Hessian-preconditioned inner products using random projections (Dasgupta & Gupta, 2003) rather than K-FAC. We also compare to the ad-hoc changes to the measurement/training loss in the influence function approximation (D-TRAK) that were shown by Zheng et al. (2024) to give improved LDS performance. Note that, the changes in D-TRAK were directly optimised for improvements in LDS scores in the diffusion modelling setting, and lack any theoretical motivation. Hence, a direct comparison for the changes proposed in this work (K-FAC Influence) is TRAK; the insights from D-TRAK are orthogonal to our work. These are the only prior works motivated by predicting the change in a model's measurements after retraining that have been applied to the general diffusion modelling setting that we are aware of. We also compare to naïvely using cosine similarity between the CLIP (Radford et al., 2021) embeddings of the training datapoints and the generated sample as a proxy for influence on the generated samples. Lastly, we report LDS results for the oracle method of "Exact Retraining," where we actually retrain a single model to predict the changes in measurements.

**LDS.** The LDS results attributing the loss and ELBO measurements are shown in Figures 2a and 2b. The LDS results attributing the marginal log-probability on dequantised data are shown in Appendix I. K-FAC Influence outperforms TRAK in all settings. K-FAC Influence using the loss measurement also outperforms the benchmark-tuned changes in D-TRAK in all settings as well. In Figures 2a, 2b and 21, we report the results for both the best damping values from a sweep (see Appendix G), as well as for "default" values following recommendations in previous work (see Appendix J.5). TRAK and D-TRAK appear to be more sensitive to tuning the damping factor than K-FAC Influence. They often don't perform at all if the damping factor is too small, and take a noticeable performance hit if the damping factor is not tuned to the problem or method (see Figures 8 and 10 in Appendix G). However, in most applications, tuning the damping factor would be infeasible, as it requires retraining the model many times over to construct an LDS benchmark, so this is a significant limitation. In contrast, for K-FAC Influence, we find that generally any sufficiently small value works reasonably well (see Figures 7 and 9).

**Retraining without top influences.** The counterfactual retraining results are shown in Figure 3 for CIFAR-2, CIFAR-10, with $2\%$ and $10\%$ of the data removed. In this evaluation, influence functions with K-FAC consistently pick more influential training examples (i.e. those which lead to a higher loss reduction) than the baselines.

## 4.1 POTENTIAL LIMITATIONS OF INFLUENCE FUNCTIONS FOR DIFFUSION MODELS

One peculiarity in the LDS results, similar to the findings in Zheng et al. (2024), is that substituting the loss measurement for the ELBO measurement when predicting changes in ELBO or the marginal log-probability actually works better than using the correct measurement (see Figure 2b "K-FAC Influence (measurement loss)").[9] To try and better understand the properties of influence functions, in this section we perform multiple ablations and report different interesting phenomena that give some insight into the challenges of using influence functions in this setting.

---

[9]Note that, unlike Zheng et al. (2024), we only change the measurement function for a proxy in the influence function approximation, keeping the Hessian approximation and training loss gradient in Equation (6) the same.

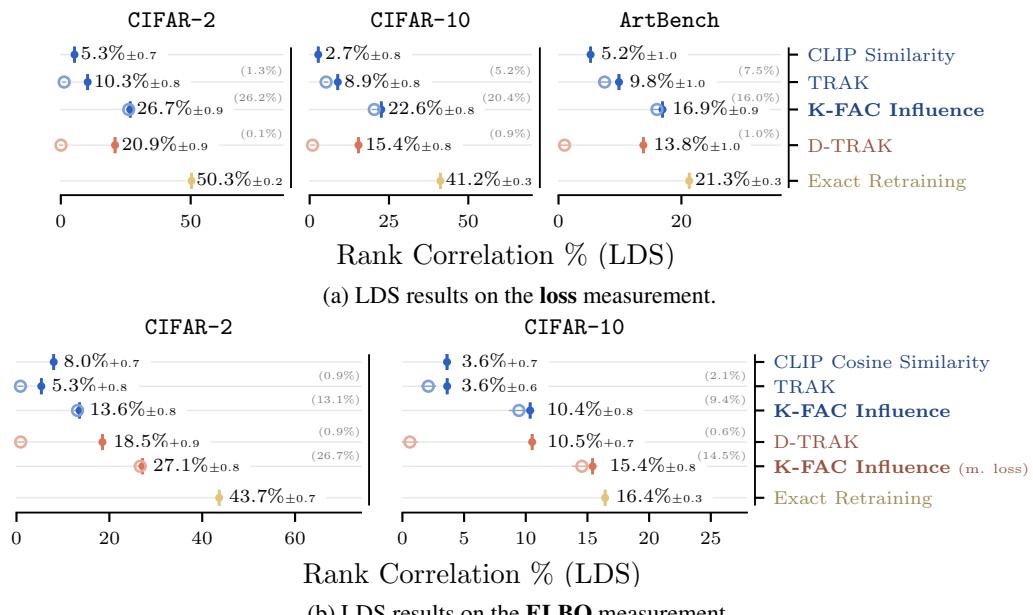

(a) LDS results on the **loss** measurement.

(b) LDS results on the **ELBO** measurement.

Figure 2: Linear Data-modelling Score (LDS) for different data attribution methods. Methods that substitute in *incorrect* measurement functions into the approximation are separated and plotted with •. Where applicable, we plot results for both the best Hessian-approximation damping value with • and a "default" damping value with ○. The numerical results are reported in black for the best damping value, and for the "default" damping value in (gray). "(m. loss)" implies that the appropriate measurement function was substituted with the loss $\ell(\theta, x)$ measurement function in the approximation. Results for the exact retraining method (oracle), are shown with •. Standard error in the LDS score estimate is indicated with '±', where the mean is taken over different generated samples $x$ on which the change in measurement is being estimated.

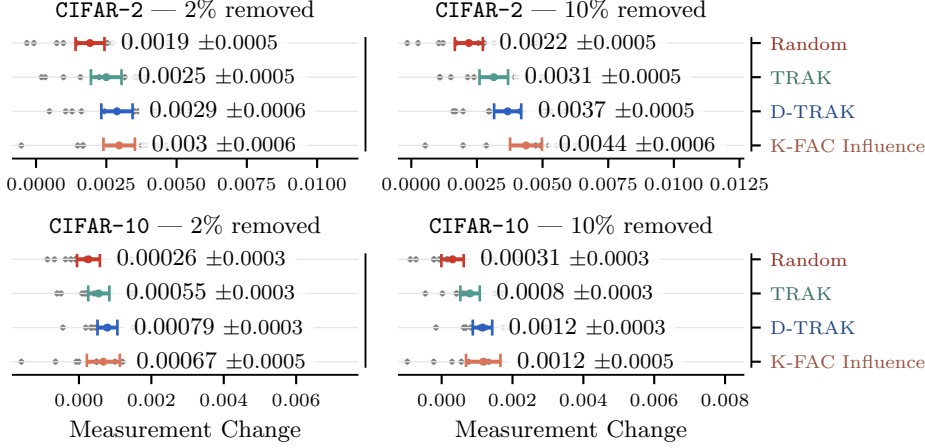

Figure 3: Changes in measurements under counterfactual retraining without top influences for the **loss** measurement. The standard error in the estimate of the mean is indicated with error bars and reported after '±', where the average is over different generated samples for which top influences are being identified.

As illustrated in Figure 20, gradients of the ELBO and training loss measurements, up to a constant scaling, consist of the same per-diffusion-timestep loss term gradients $\nabla_\theta \ell_t(\theta, x)$, but with a different weighting. To try and break-down why approximating the change in ELBO with the training loss measurement gives higher LDS scores, we first look at predicting the change in the per-diffusion-

timestep losses $\ell_t$ while substituting *different* per-diffusion-timestep losses into the K-FAC influence approximation. The results are shown in Figure 13, leading to the following observation:

> **Observation 1** *Higher-timestep losses $\ell_t(\theta, x)$ act as better proxies for lower-timestep losses.*

More specifically, changes in losses $\ell_t$ can in general be well approximated by substituting measurements $\ell_{t'}$ into the influence approximation with $t' > t$. In some cases, using the incorrect timestep $t' > t$ even results in significantly better LDS scores than the correct timestep $t' = t$.

Based on Observation 1, it is clear that influence function-based approximations have limitations when being applied to predict the numerical change in loss measurements. We observe another pattern in how they can fail:

> **Observation 2** *Influence functions predict both positive and negative influence on loss, but, in practice, removing data points predominantly increases loss.*

We show in Figures 17 and 18 that influence functions tend to overestimate how often removal of a group data points will lead to improvements in loss on a generated sample (both for aggregate diffusion training loss in Section 2.1, and the per-diffusion-timestep loss in Equation (2)).

Lastly, although we have argued for estimating the effect that removing training datapoints has on the model's marginal distribution, one property of diffusion models complicates the usefulness of pursuing this goal in practice for models trained on large amounts of data:

> **Observation 3** *For sufficiently large training set sizes, the diffusion model's marginal probability distribution is close to constant (on generated samples), irrespective of which examples were removed from the training data.*

As illustrated in Figure 19a, the exact marginal log-probability measurement is close to constant for any given sample generated from the model, no matter which $50\%$ subset of the training data is removed if the resulting training dataset is large enough. In particular, it is extremely rare that one sample is more likely to be generated than another by one model, and is less likely to be generated than another by a different model trained on a different random subset of the data. Our observation mirrors that of Kadkhodaie et al. (2024) who found that, if diffusion models are trained on non-overlapping subsets of data of sufficient size, they generate near-identical samples when fed with the random seed. We find this observation holds not just for the generated samples, but for the marginal log-probability density, the ELBO and the training loss measurements as well (see Figures 19b and 19c).

## 5 Discussion

In this work, we extended the influence functions approach to the diffusion modelling setting, and showed different ways in which the GGN Hessian approximation can be formulated. Our proposed method with recommended design choices improves performance compared to existing techniques across various data attribution evaluation metrics. Nonetheless, experimentally, we are met with two contrasting findings: on the one hand, influence functions in the diffusion modelling setting appear to be able to identify important influences. The surfaced influential examples do significantly impact the training loss when retraining the model without them (Figure 3), and they appear perceptually very relevant to the generated samples. On the other hand, they fall short of accurately predicting the numerical changes in measurements after retraining. This appears to be especially the case for measurement functions we would argue are most relevant in the image generative modelling setting – proxies for marginal probability of sampling a particular example (Section 4.1).

Despite these shortcomings, influence functions can still offer valuable insights: they can serve as a useful exploratory tool for understanding model behaviour in a diffusion modelling context, and can help guide data curation, identifying examples most responsible for certain behaviours. To make them useful in settings where numerical accuracy in the predicted behaviour after retraining is required, such as copyright infringement, we believe more work is required into **1)** finding better proxies for marginal probability, and **2)** even further improving the influence functions approximation.

ACKNOWLEDGMENTS

We thank Jihao Andreas Lin for useful discussions on compression, and help with implementation of quantisation and SVD compression. We also thank Kristian Georgiev for sharing with us the diffusion model weights used for analysis in Georgiev et al. (2023), and Felix Dangel for help and feedback on implementing extensions to the `curvlinops` (Dangel et al., 2025) package used for the experiments in this paper. Richard E. Turner is supported by the EPSRC Probabilistic AI Hub (EP/Y028783/1).

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

## A  DERIVATION OF INFLUENCE FUNCTIONS

In this section, we state the implicit function theorem (Appendix A.1). Then, in Appendix A.2, we introduce the details of how it can be applied in the context of a loss function $\mathcal{L}(\varepsilon, \boldsymbol{\theta})$ parameterised by a continuous hyperparameter $\varepsilon$ (which is, e.g., controlling how down-weighted the loss terms on some examples are, as in Section 2.2).

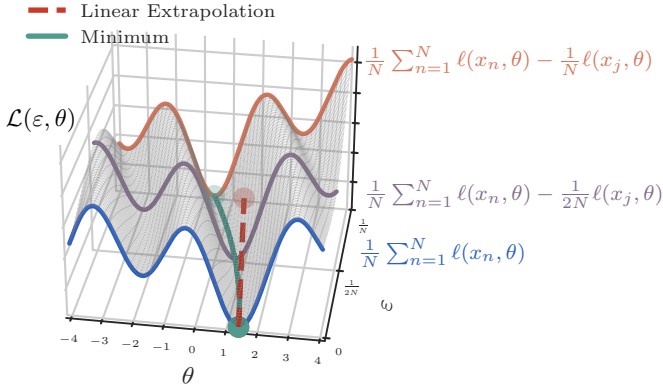

Figure 4: Illustration of the influence function approximation for a 1-dimensional parameter space $\theta \in \mathbb{R}$. Influence funcitons consider the extended loss landscape $\mathcal{L}(\varepsilon, \theta) \stackrel{\text{def}}{=} \frac{1}{N} \sum_{n=1}^{N} \ell(x_n, \theta) - \varepsilon \ell(x_j, \theta)$, where the loss $\ell(x_j, \theta)$ for some datapoint $x_j$ (alternatively, group of datapoints) is down-weighted by $\varepsilon$. By linearly extrapolating how the optimal set of parameters $\theta$ would change around $\varepsilon = 0$ (•), we can predicted how the optimal parameters would change when the term $\ell(x_j, \theta)$ is fully removed from the loss (•).

### A.1  IMPLICIT FUNCTION THEOREM

**Theorem 1 (Implicit Function Theorem (Krantz & Parks, 2003))** *Let $F : \mathbb{R}^n \times \mathbb{R}^m \to \mathbb{R}^m$ be a continuously differentiable function, and let $\mathbb{R}^n \times \mathbb{R}^m$ have coordinates $(\mathbf{x}, \mathbf{y})$. Fix a point $(\mathbf{a}, \mathbf{b}) = (a_1, \ldots, a_n, b_1, \ldots, b_m)$ with $F(\mathbf{a}, \mathbf{b}) = \mathbf{0}$, where $\mathbf{0} \in \mathbb{R}^m$ is the zero vector. If the Jacobian matrix $\nabla_{\mathbf{y}} F(\mathbf{a}, \mathbf{b}) \in \mathbb{R}^{m \times m}$ of $\mathbf{y} \mapsto F(\mathbf{a}, \mathbf{y})$*

$$[\nabla_{\mathbf{y}} F(\mathbf{a}, \mathbf{b})]_{ij} = \frac{\partial F_i}{\partial y_j}(\mathbf{a}, \mathbf{b}),$$

*is invertible, then there exists an open set $U \subset \mathbb{R}^n$ containing $\mathbf{a}$ such that there exists a unique function $g : U \to \mathbb{R}^m$ such that $g(\mathbf{a}) = \mathbf{b}$, and $F(\mathbf{x}, g(\mathbf{x})) = \mathbf{0}$ for all $\mathbf{x} \in U$. Moreover, $g$ is continuously differentiable.*

**Remark 1 (Derivative of the implicit function)** *Denoting the Jacobian matrix of $\mathbf{x} \mapsto F(\mathbf{x}, \mathbf{y})$ as:*

$$[\nabla_{\mathbf{x}} F(\mathbf{x}, \mathbf{y})]_{ij} = \frac{\partial F_i}{\partial x_j}(\mathbf{x}, \mathbf{y}),$$

*the derivative $\frac{\partial \mathbf{g}}{\partial \mathbf{x}} : U \to \mathbb{R}^{m \times n}$ of $g : U \to \mathbb{R}^m$ in Theorem 1 can be written as:*

$$\frac{\partial g(\mathbf{x})}{\partial \mathbf{x}} = -\left[\nabla_{\mathbf{y}} F(\mathbf{x}, g(\mathbf{x}))\right]^{-1} \nabla_{\mathbf{x}} F(\mathbf{x}, g(\mathbf{x})). \tag{14}$$

*This can readily be seen by noting that, for $\mathbf{x} \in U$:*

$$F(\mathbf{x}', g(\mathbf{x}')) = \mathbf{0} \quad \forall \mathbf{x}' \in U \qquad \Rightarrow \qquad \frac{dF(\mathbf{x}, g(\mathbf{x}))}{d\mathbf{x}} = \mathbf{0}.$$

*Hence, since $g$ is differentiable, we can apply the chain rule of differentiation to get:*

$$\mathbf{0} = \frac{dF(\mathbf{x}, g(\mathbf{x}))}{d\mathbf{x}} = \nabla_{\mathbf{x}} F(\mathbf{x}, g(\mathbf{x})) + \nabla_{\mathbf{y}} F(\mathbf{x}, g(\mathbf{x})) \frac{\partial g(\mathbf{x})}{\partial \mathbf{x}}.$$

*Rearranging gives equation Equation (14).*

## A.2 APPLYING THE IMPLICIT FUNCTION THEOREM TO QUANTIFY THE CHANGE IN THE OPTIMUM OF A LOSS

Consider a loss function $\mathcal{L} : \mathbb{R}^n \times \mathbb{R}^m \to \mathbb{R}$ that depends on some hyperparameter $\varepsilon \in \mathbb{R}^n$ (in Section 2.2, this was the scalar by which certain loss terms were down-weighted) and some parameters $\boldsymbol{\theta} \in \mathbb{R}^m$. At the minimum of the loss function $\mathcal{L}(\varepsilon, \boldsymbol{\theta})$, the derivative with respect to the parameters $\boldsymbol{\theta}$ will be zero. Hence, assuming that the loss function is twice continuously differentiable (hence $\frac{\partial L}{\partial \varepsilon}$ is continuously differentiable), and assuming that for some $\varepsilon' \in \mathbb{R}^n$ we have a set of parameters $\boldsymbol{\theta}^\star$ such that $\frac{\partial \mathcal{L}}{\partial \varepsilon}(\varepsilon', \boldsymbol{\theta}^\star) = \mathbf{0}$ and the Hessian $\frac{\partial^2 \mathcal{L}}{\partial \boldsymbol{\theta}^2}(\varepsilon', \boldsymbol{\theta}^\star)$ is invertible, we can apply the implicit function theorem to the derivative of the loss function $\frac{\partial \mathcal{L}}{\partial \varepsilon} : \mathbb{R}^n \times \mathbb{R}^m \to \mathbb{R}^m$, to get the existence of a continuously differentiable function $g$ such that $\frac{\partial \mathcal{L}}{\partial \varepsilon}(\varepsilon, g(\varepsilon)) = \mathbf{0}$ for $\varepsilon$ in some neighbourhood of $\varepsilon'$.

Now $g(\varepsilon)$ might not necessarily be a minimum of $\boldsymbol{\theta} \mapsto \mathcal{L}(\varepsilon, \boldsymbol{\theta})$. However, by making the further assumption that $\mathcal{L}$ is strictly convex we can ensure that whenever $\frac{\partial \mathcal{L}}{\partial \boldsymbol{\theta}}(\varepsilon, \boldsymbol{\theta}) = \mathbf{0}$, $\boldsymbol{\theta}$ is a unique minimum, and so $g(\varepsilon)$ represents the change in the minimum as we vary $\varepsilon$. This is summarised in the lemma below:

**Lemma 1** *Let $\mathcal{L} : \mathbb{R}^n \times \mathbb{R}^m \to \mathbb{R}$ be a twice continuously differentiable function, with coordinates denoted by $(\varepsilon, \boldsymbol{\theta}) \in \mathbb{R}^n \times \mathbb{R}^m$, such that $\boldsymbol{\theta} \mapsto \mathcal{L}(\varepsilon, \boldsymbol{\theta})$ is strictly convex $\forall \varepsilon \in \mathbb{R}^n$. Fix a point $(\varepsilon', \boldsymbol{\theta}^\star)$ such that $\frac{\partial \mathcal{L}}{\partial \boldsymbol{\theta}}(\varepsilon', \boldsymbol{\theta}^\star) = \mathbf{0}$. Then, by the Implicit Function Theorem applied to $\frac{\partial \mathcal{L}}{\partial \boldsymbol{\theta}}$, there exists an open set $U \subset \mathbb{R}^n$ containing $\boldsymbol{\theta}^\star$ such that there exists a unique function $g : U \to \mathbb{R}^m$ such that $g(\varepsilon') = \boldsymbol{\theta}^\star$, and $g(\varepsilon)$ is the unique minimum of $\boldsymbol{\theta} \mapsto \mathcal{L}(\varepsilon, \boldsymbol{\theta})$ for all $\varepsilon \in U$. Moreover, $g$ is continuously differentiable with derivative:*

$$\frac{\partial g(\varepsilon)}{\partial \varepsilon} = -\left[\frac{\partial^2 \mathcal{L}}{\partial \boldsymbol{\theta}^2}(\varepsilon, g(\varepsilon))\right]^{-1} \frac{\partial^2 \mathcal{L}}{\partial \varepsilon \partial \boldsymbol{\theta}}(\varepsilon, g(\varepsilon)) \tag{15}$$

**Remark 2** *For a loss function $\mathcal{L} : \mathbb{R} \times \mathbb{R}^m$ of the form $\mathcal{L}(\varepsilon, \boldsymbol{\theta}) = \mathcal{L}_1(\boldsymbol{\theta}) + \varepsilon \mathcal{L}_2(\boldsymbol{\theta})$ (such as that in Equation (4)), $\frac{\partial^2 \mathcal{L}}{\partial \varepsilon \partial \boldsymbol{\theta}}(\varepsilon, g(\varepsilon))$ in the equation above simplifies to:*

$$\frac{\partial^2 \mathcal{L}}{\partial \varepsilon \partial \boldsymbol{\theta}}(\varepsilon, g(\varepsilon)) = \frac{\partial \mathcal{L}_2}{\partial \boldsymbol{\theta}}(g(\varepsilon)) \tag{16}$$

The above lemma and remark give the result in Equation (5). Namely, in section 2.2:

$$\mathcal{L}(\varepsilon, \boldsymbol{\theta}) = \underbrace{\frac{1}{N}\sum_{i=1}^{N} \ell(\boldsymbol{\theta}, x_i)}_{\mathcal{L}_1} - \overbrace{\frac{1}{M}\sum_{j=1}^{M} \ell(\boldsymbol{\theta}, x_{i_j})\,\varepsilon}^{\mathcal{L}_2} \qquad \overset{\text{eq. (16)}}{\Longrightarrow} \qquad \frac{\partial^2 \mathcal{L}}{\partial \varepsilon \partial \boldsymbol{\theta}} = -\frac{1}{M}\sum_{j=1}^{M}\frac{\partial}{\partial \boldsymbol{\theta}}\ell(\boldsymbol{\theta}, x_{i_j})$$

$$\overset{\text{eq. (15)}}{\Longrightarrow} \qquad \frac{\partial g(\varepsilon)}{\partial \varepsilon} = \left[\frac{\partial^2 \mathcal{L}}{\partial \boldsymbol{\theta}^2}(\varepsilon, g(\varepsilon))\right]^{-1} \frac{1}{M}\sum_{j=1}^{M}\frac{\partial}{\partial \boldsymbol{\theta}}\ell(\boldsymbol{\theta}, x_{i_j})$$

## B DERIVATION OF THE FISHER "GGN" FORMULATION FOR DIFFUSION MODELS

As discussed in Section 2.3 partitioning the function $\boldsymbol{\theta} \mapsto \|\epsilon^{(t)} - \epsilon_\theta^t(x^{(t)})\|^2$ into the model output $\boldsymbol{\theta} \mapsto \epsilon_\theta^t(x^{(t)})$ and the $\ell_2$ loss function is a natural choice and results in

$$\text{GGN}_{\mathcal{D}}^{\texttt{model}}(\theta)$$

$$= \frac{1}{N}\sum_{n=1}^{N} \mathbb{E}_{\tilde{t}}\left[\mathbb{E}_{x^{(\tilde{t})}, \epsilon^{(\tilde{t})}}\left[\nabla_\theta^\intercal \epsilon_\theta^{\tilde{t}}\left(x^{(\tilde{t})}\right) \nabla_{\epsilon_\theta^{\tilde{t}}(x^{(\tilde{t})})}^2 \left\|\epsilon^{(\tilde{t})} - \epsilon_\theta^{\tilde{t}}\left(x^{(\tilde{t})}\right)\right\|^2 \nabla_\theta \epsilon_\theta^{\tilde{t}}\left(x^{(\tilde{t})}\right)\right]\right]$$

$$= \frac{2}{N}\sum_{n=1}^{N} \mathbb{E}_{\tilde{t}}\left[\mathbb{E}_{x^{(\tilde{t})}, \epsilon^{(\tilde{t})}}\left[\nabla_\theta^\intercal \epsilon_\theta^{\tilde{t}}\left(x^{(\tilde{t})}\right) I \nabla_\theta \epsilon_\theta^{\tilde{t}}\left(x^{(\tilde{t})}\right)\right]\right]. \tag{17}$$

Note that we used

$$\frac{1}{2} \nabla^2_{\epsilon^{\tilde{t}}_\theta(x^{(\tilde{t})})} \left\| \epsilon^{(\tilde{t})} - \epsilon^{\tilde{t}}_\theta \left( x^{(\tilde{t})} \right) \right\|^2 = I.$$

We can substitute $I$ with $\mathbb{E}_\eta \left[ \eta \eta^\intercal \right]$ for any random vector $\eta$ with identity second moment. This allows us to rewrite the expression for the GGN in Equation (17) as

$$\mathrm{GGN}^{\texttt{model}}_\mathcal{D}(\theta) = \frac{2}{N} \sum_{n=1}^N \mathbb{E}_{\tilde{t}} \left[ \mathbb{E}_{x^{(\tilde{t})}, \epsilon^{(\tilde{t})}, \eta} \left[ \nabla^\intercal_\theta \epsilon^{\tilde{t}}_\theta \left( x^{(\tilde{t})} \right) \eta \eta^\top \nabla_\theta \epsilon^{\tilde{t}}_\theta \left( x^{(\tilde{t})} \right) \right] \right].$$

where the term $\nabla^\intercal_\theta \epsilon^{\tilde{t}}_\theta \left( x^{(\tilde{t})} \right) \eta \in \mathbb{R}^{d_{\texttt{param}}}$, which is a vector-Jacobian product, can be efficiently evaluated at roughly the cost of a single backward pass. For example, it can be rewritten in a manner lending itself to implementation in a standard autodiff library as:

$$\nabla^\intercal_\theta \epsilon^{\tilde{t}}_\theta \left( x^{(\tilde{t})} \right) \eta = \frac{1}{2} \nabla_{\epsilon^{\tilde{t}}_\theta(x^{(\tilde{t})})} \left\| \epsilon_{\texttt{mod}} - \epsilon^{\tilde{t}}_\theta \left( x^{(\tilde{t})} \right) \right\|^2 \qquad \epsilon_{\texttt{mod}} := \texttt{stopgrad} \left[ \epsilon^{\tilde{t}}_\theta \left( x^{(\tilde{t})} \right) \right] + \eta,$$

where $\texttt{stopgrad}$ denotes the common stopping of passing of the adjoint through a certain operation common in autodiff frameworks.

## C  (E)K-FAC FOR DIFFUSION MODELS

### C.1  BACKGROUND: KRONECKER-FACTORED APPROXIMATE CURVATURE

Kronecker-Factored Approximate Curvature (Heskes, 2000; Martens & Grosse, 2015; Botev et al., 2017, K-FAC) is typically used as a layer-wise block-diagonal approximation of the Fisher or GGN of a neural network. Each layer-wise block can be written as a Kronecker product, hence the name.

We will first describe the K-FAC approximation for a simple linear layer for a deep neural network for a standard regression or classification setting. We assume a loss function $\frac{1}{N} \sum_{n=1}^N \ell(y_n, f_\theta(x_n))$ where $f_\theta$ is a neural network parametrised by $\theta$, $\mathcal{D} = \{x_n, y_n\}_{n=1}^N$ is the dataset with inputs $x_n$ and labels $y_n$, and $\ell(\cdot, \cdot)$ is a loss function like the cross-entropy or mean square error. To derive the K-FAC approximation for the parameters of a linear layer with weight matrix $W_l$[10], we first note that we can write the GGN block for the flattened parameters $\theta_l = \mathrm{vec}(W_l)$ as

$$\mathrm{GGN}_\mathcal{D}(\theta_l) = \frac{1}{N} \sum_{n=1}^N \nabla^\intercal_{\theta_l} f_\theta(x_n) \left( \nabla^2_{f_\theta} \ell(y_n, f_\theta(x_n)) \right) \nabla_{\theta_l} f_\theta(x_n); \tag{18}$$

here we choose the split from $\mathrm{GGN}^{\mathrm{model}}$ in Equation (7), but the derivation also follows similarly for other splits. Given that $\nabla^\intercal_{\theta_l} f_\theta(x_n) = a_n^{(l)} \otimes g_n^{(l)}$, where $a_n^{(l)}$ is the input to the $l$th layer for the $n$th example and $g_n^{(l)}$ is the transposed Jacobian of the neural network output w.r.t. to the output of the $l$th layer for the $n$th example, we have

$$\mathrm{GGN}_\mathcal{D}(\theta_l) = \frac{1}{N} \sum_{n=1}^N \left( a_n^{(l)} \otimes g_n^{(l)} \right) \left( \nabla^2_{f_\theta} \ell(y_n, f_\theta(x_n)) \right) \left( a_n^{(l)} \otimes g_n^{(l)} \right)^\intercal \tag{19}$$

$$= \frac{1}{N} \sum_{n=1}^N \left( a_n^{(l)} a_n^{(l)\intercal} \right) \otimes \left( g_n^{(l)} \left( \nabla^2_{f_\theta} \ell(y_n, f_\theta(x_n)) \right) g_n^{(l)\intercal} \right). \tag{20}$$

K-FAC is now approximating this sum of Kronecker products with a Kronecker product of sums, i.e.

$$\mathrm{GGN}_\mathcal{D}(\theta_l) \approx \frac{1}{N^2} \left( \sum_{n=1}^N a_n^{(l)} a_n^{(l)\intercal} \right) \otimes \left( \sum_{n=1}^N g_n^{(l)} \left( \nabla^2_{f_\theta} \ell(y_n, f_\theta(x_n)) \right) g_n^{(l)\intercal} \right). \tag{21}$$

This approximation becomes an equality in the trivial case of $N = 1$ or for simple settings of deep linear networks with mean square error loss function (Bernacchia et al., 2018). After noticing

---

[10]A potential bias vector can be absorbed into the weight matrix.

that the Hessian $\nabla^2_{f_\theta} \ell(y_n, f_\theta(x_n))$ is the identity matrix for the mean square error loss, the K-FAC formulation for diffusion models in Equation (12) can now be related to this derivation – the only difference is the expectations from the diffusion modelling objective.

Note that this derivation assumed a simple linear layer. However, common architectures used for diffusion models consist of different layer types as well, such as convolutional layers and attention. As mentioned in Section 3.1.2, K-FAC can be more generally formulated for all linear layers with weight sharing (Eschenhagen et al., 2023).

First, note that the core building blocks of common neural network architectures can be expressed as linear layers with weight sharing. If a linear layer without weight sharing can be thought of a weight matrix $W \in \mathbb{R}^{d_{\text{out}} \times d_{\text{in}}}$ that is applied to an input vector $x \in \mathbb{R}^{d_{\text{in}}}$, a linear weight sharing layer is applying the transposed weight matrix to right of an input matrix $X \in \mathbb{R}^{M \times d_{\text{in}}}$, i.e. $XW^\intercal$. This can be thought of a regular linear layer that is shared across the additional input dimension of size $M$. For example, the weight matrices in the attention mechanism are shared across tokens, the kernel in convolutions is shared across the spatial dimensions, and in a graph neural network layer the weights might be shared across nodes or edges; see Section 2.2 in Eschenhagen et al. (2023) for a more in-depth explanation of these examples.

Given this definition of linear weight sharing layers, we can identify two different settings in which they are used. In the *expand* setting, we have $R \times N$ loss terms for a dataset with $N$ data points; we have a loss of the form $\frac{1}{NR} \sum_{n=1}^{N} \sum_{r=1}^{R} \ell(y_{n,r}, f_\theta(X_n)_r)$. The additional $R$ loss terms can sometimes exactly correspond to the weight sharing dimension of size $M$, i.e., $R = M$; for example, when the weight sharing dimension corresponds to tokens in a sequence as it is the case for linear layers within attention. In contrast, in the *reduce* setting, we have a loss of the form $\frac{1}{N} \sum_{n=1}^{N} \ell(y_n, f_\theta(X_n))$.[11] These two settings can now be used to motivate two different flavours of the K-FAC approximation.

The first flavour, **K-FAC-expand**, is defined as

$$\text{GGN}_{\mathcal{D}}(\theta_l) \approx \frac{1}{c_{\text{xpnd}}} \left( \sum_{n=1}^{N} \sum_{m=1}^{M} a_{n,m}^{(l)} a_{n,m}^{(l)\intercal} \right) \otimes \left( \sum_{n=1}^{N} \sum_{m=1}^{M} \left( \sum_{r=1}^{R} g_{n,r,m}^{(l)} H_{n,r}^{\frac{1}{2}} \right) \left( \sum_{r=1}^{R} H_{n,r}^{\frac{1}{2}} g_{n,r,m}^{(l)\intercal} \right) \right),$$
(22)

where $c_{\text{xpnd}} = N^2 RM$, $a_{n,m}^{(l)}$ is the $m$th row of the input to the $l$th layer for the $n$th example, $H_{n,r} = \nabla^2_{f_\theta} \ell(y_{n,r}, f_\theta(X_n)_r)$, and $g_{n,r,m}^{(l)}$ is the transposed Jacobian of the $r$th row of the matrix output of the neural network w.r.t. the $m$th row of the output matrix of the $l$th layer for the $n$th example. K-FAC-expand is motivated by the expand setting in the sense that for deep linear networks with a mean square error as the loss function, K-FAC-expand is exactly equal to the layer-wise block-diagonal of the GGN. For convolutions, K-FAC expand corresponds to the K-FAC approximation derived in Grosse & Martens (2016) which has also been used for attention before (Zhang et al., 2019; Pauloski et al., 2021; Osawa et al., 2022).

The second variation, **K-FAC-reduce**, is defined as

$$\text{GGN}_{\mathcal{D}}(\theta_l) \approx \frac{1}{c_{\text{rdc}}} \left( \sum_{n=1}^{N} \left( \sum_{m=1}^{M} a_{n,m}^{(l)} \right) \left( \sum_{m=1}^{M} a_{n,m}^{(l)\intercal} \right) \right) \otimes \left( \sum_{n=1}^{N} \left( \sum_{m=1}^{M} g_{n,m}^{(l)} \right) H_n \left( \sum_{m=1}^{M} g_{n,m}^{(l)\intercal} \right) \right),$$
(23)

with $c_{\text{rdc}} = (NM)^2$. Analogously to K-FAC-expand in the expand setting, in the reduce setting, K-FAC reduce is exactly equal to the layer-wise block-diagonal GGN for a deep linear network with mean square error loss and a scaled sum as the reduction function. With reduction function we refer to the function that is used to reduce the weight-sharing dimension in the forward pass of the model, e.g. average pooling to reduce the spatial dimension in a convolutional neural network. Similar approximations have also been proposed in a different context (Tang et al., 2021; Immer et al., 2022).

Although each setting motivates a corresponding K-FAC approximation in the sense described above, we can apply either K-FAC approximation in each setting. Also, note that while we only explicitly consider a single weight sharing dimension, there can in principle be an arbitrary number of them and we can choose between K-FAC-expand and K-FAC-reduce for each of them independently.

---

[11]In principal, the input to the neural network does not necessarily have to have a weight-sharing dimension, even when we the model contains linear weigh-sharing layers; this holds for the expand and the reduce setting.

## C.2 DERIVATION OF K-FAC FOR DIFFUSION MODELS

We will now explicitly derive K-FAC for diffusion models while also taking weight sharing into account. As described above, we will derive K-FAC-expand as well as K-FAC-reduce for $F_{\mathcal{D}}$ (Equation (10)), or equivalently, $\text{GGN}_{\mathcal{D}}^{\text{model}}$ (Equation (9)). Note that once we have derived K-FAC, we can immediately apply an eigenvalue-correction to get EK-FAC (George et al., 2018).

Let $g(x_n^{(\tilde{t})}, \epsilon^{(\tilde{t})}) = \nabla_\theta \|\epsilon^{(\tilde{t})} - \epsilon_\theta(x_n^{(\tilde{t})})\|^2 = \sum_{r=1}^{R} \nabla_\theta (\epsilon_r^{(\tilde{t})} - \epsilon_\theta(x_n^{(\tilde{t})})_r)^2 = \sum_{r=1}^{R} \sum_{m=1}^{M} a_m^{(l)} \otimes b_{r,m}^{(l)}$, where $a_m^{(l)}$ is the $m$th row of the input to the $l$th layer, $b_{r,m}^{(l)}$ is the gradient of $(\epsilon_r^{(\tilde{t})} - \epsilon_\theta(x_n^{(\tilde{t})})_r)^2$ w.r.t. the $m$th row of the $l$th layer output, $M$ is the size of the weight sharing dimension of the $l$th layers, and $R$ is the size of the output dimension, e.g., height × width × number of channels for image data. We have

$$F_{\mathcal{D}}(\theta_l) = \mathbb{E}_{x_n}\left[\mathbb{E}_{\tilde{t}}\left[\mathbb{E}_{x_n^{(\tilde{t})}, \epsilon^{(\tilde{t})}, \epsilon_{\text{mod}}^{(\tilde{t})}}\left[g\left(x_n^{(\tilde{t})}, \epsilon_{\text{mod}}^{(\tilde{t})}\right) g\left(x_n^{(\tilde{t})}, \epsilon_{\text{mod}}^{(\tilde{t})}\right)^T\right]\right]\right] \tag{24}$$

$$= \mathbb{E}_{x_n}\left[\mathbb{E}_{\tilde{t}}\left[\mathbb{E}_{x_n^{(\tilde{t})}, \epsilon^{(\tilde{t})}, \epsilon_{\text{mod}}^{(\tilde{t})}}\left[\left(\sum_{r=1}^{R}\sum_{m=1}^{M} a_m^{(l)} \otimes b_{r,m}^{(l)}\right)\left(\sum_{r=1}^{R}\sum_{m=1}^{M} a_m^{(l)} \otimes b_{r,m}^{(l)}\right)^T\right]\right]\right] \tag{25}$$

$$= \mathbb{E}_{x_n}\left[\mathbb{E}_{\tilde{t}}\left[\mathbb{E}_{x_n^{(\tilde{t})}, \epsilon^{(\tilde{t})}, \epsilon_{\text{mod}}^{(\tilde{t})}}\left[\left(\sum_{m=1}^{M} a_m^{(l)} \otimes \underbrace{\sum_{r=1}^{R} b_{r,m}^{(l)}}_{\hat{b}_m^{(l)} :=}\right)\left(\sum_{m=1}^{M} a_m^{(l)} \otimes \sum_{r=1}^{R} b_{r,m}^{(l)}\right)^T\right]\right]\right]. \tag{26}$$

$$\tag{27}$$

First, we can use K-FAC-expand, which is exact for the case of a (deep) linear network:

$$F_{\mathcal{D}}(\theta_l) \stackrel{\text{expand}}{\approx} \mathbb{E}_{x_n}\left[\mathbb{E}_{\tilde{t}}\left[\mathbb{E}_{x_n^{(\tilde{t})}, \epsilon^{(\tilde{t})}, \epsilon_{\text{mod}}^{(\tilde{t})}}\left[\sum_{m=1}^{M}\left(a_m^{(l)} a_m^{(l)\mathsf{T}} \otimes \hat{b}_m^{(l)} \hat{b}_m^{(l)\mathsf{T}}\right)\right]\right]\right] \tag{28}$$

$$\approx \frac{1}{M}\,\mathbb{E}_{x_n}\left[\mathbb{E}_{\tilde{t}}\left[\mathbb{E}_{x_n^{(\tilde{t})}, \epsilon^{(\tilde{t})}}\left[\sum_{m=1}^{M} a_m^{(l)} a_m^{(l)\mathsf{T}}\right]\right]\right] \otimes \mathbb{E}_{x_n}\left[\mathbb{E}_{\tilde{t}}\left[\mathbb{E}_{x_n^{(\tilde{t})}, \epsilon^{(\tilde{t})}, \epsilon_{\text{mod}}^{(\tilde{t})}}\left[\sum_{m=1}^{M} \hat{b}_m^{(l)} \hat{b}_m^{(l)\mathsf{T}}\right]\right]\right] \tag{29}$$

$$\tag{30}$$

Alternatively, we can use K-FAC-reduce:

$$F_{\mathcal{D}}(\theta_l) \stackrel{\text{reduce}}{\approx} \frac{1}{M^2}\,\mathbb{E}_{x_n}\left[\mathbb{E}_{\tilde{t}}\left[\mathbb{E}_{x_n^{(\tilde{t})}, \epsilon^{(\tilde{t})}, \epsilon_{\text{mod}}^{(\tilde{t})}}\left[\left(\hat{a}^{(l)} \otimes \hat{b}^{(l)}\right)\left(\hat{a}^{(l)} \otimes \hat{b}^{(l)}\right)^\mathsf{T}\right]\right]\right] \tag{31}$$

$$= \frac{1}{M^2}\,\mathbb{E}_{x_n}\left[\mathbb{E}_{\tilde{t}}\left[\mathbb{E}_{x_n^{(\tilde{t})}, \epsilon^{(\tilde{t})}, \epsilon_{\text{mod}}^{(\tilde{t})}}\left[\hat{a}^{(l)} \hat{a}^{(l)\mathsf{T}} \otimes \hat{b}^{(l)} \hat{b}^{(l)\mathsf{T}}\right]\right]\right] \tag{32}$$

$$\approx \frac{1}{M^2}\,\mathbb{E}_{x_n}\left[\mathbb{E}_{\tilde{t}}\left[\mathbb{E}_{x_n^{(\tilde{t})}, \epsilon^{(\tilde{t})}}\left[\hat{a}^{(l)} \hat{a}^{(l)\mathsf{T}}\right]\right]\right] \otimes \mathbb{E}_{x_n}\left[\mathbb{E}_{\tilde{t}}\left[\mathbb{E}_{x_n^{(\tilde{t})}, \epsilon^{(\tilde{t})}, \epsilon_{\text{mod}}^{(\tilde{t})}}\left[\hat{b}^{(l)} \hat{b}^{(l)\mathsf{T}}\right]\right]\right], \tag{33}$$

with $\hat{a}^{(l)} = \sum_{m=1}^{M} a_m^{(l)}$ and $\hat{b}^{(l)} = \sum_{m=1}^{M} \hat{b}_m^{(l)} = \sum_{m=1}^{M} \sum_{r=1}^{R} b_{r,m}^{(l)}$.

To approximate $\text{GGN}_{\mathcal{D}}^{\text{loss}}$ (Equation (11)) instead of $\text{GGN}_{\mathcal{D}}^{\text{model}}$ with K-FAC, we only have to make minimal adjustments to the derivation above. Importantly, the expectation over $\epsilon_{\text{mod}}^{(\tilde{t})}$ is replaced by an expectation over $\epsilon^{(\tilde{t})}$, i.e. we use targets sampled from the same distribution as in training instead of the model's output distribution. Notably, for K-FAC-expand the approximation will be the same — ignoring the distribution over targets — as for $\text{GGN}_{\mathcal{D}}^{\text{model}}$, since we have to move all expectations and sums in the definition of $\text{GGN}_{\mathcal{D}}^{\text{loss}}$ to the front to be able to proceed (the approximation denoted with $\stackrel{\text{expand}}{\approx}$ above). However, for K-FAC-reduce the resulting approximation will be different:

$$\text{GGN}_{\mathcal{D}}^{\text{loss}}(\theta_l) \stackrel{\text{reduce}}{\approx} \frac{1}{M^2}\,\mathbb{E}_{x_n}\left[\bar{a}^{(l)} \bar{a}^{(l)\mathsf{T}}\right] \otimes \mathbb{E}_{x_n}\left[\bar{b}^{(l)} \bar{b}^{(l)\mathsf{T}}\right], \tag{34}$$

with $\bar{a}^{(l)} = \mathbb{E}_{\tilde{t}}\left[\mathbb{E}_{x_n^{(\tilde{t})}, \epsilon^{(\tilde{t})}}\left[\hat{a}^{(l)}\right]\right]$ and $\bar{b}^{(l)} = \mathbb{E}_{\tilde{t}}\left[\mathbb{E}_{x_n^{(\tilde{t})}, \epsilon^{(\tilde{t})}, \epsilon^{(\tilde{t})}}\left[\hat{b}^{(l)}\right]\right]$.

## C.3 EMPIRICAL ABLATION

Here, we explore the impact of the Hessian approximation design choices discussed in Section 3.1 and Appendix C.2. We use **K-FAC** or **EK-FAC** to approximate either the $\mathrm{GGN}_{\mathcal{D}}^{\mathrm{model}}$ in Equation (10) or the $\mathrm{GGN}_{\mathcal{D}}^{\mathrm{loss}}$ in Equation (11). We also compare the **"expand"** and the **"reduce"** variants generally introduced in Appendix C.1 and derived for our setting in Appendix C.2.

Firstly, we find that the better-motivated "MC-Fisher" estimator of $\mathrm{GGN}^{\mathrm{model}}$ in Equation (9) does indeed perform better than the "empirical Fisher" in Equation (11) used in TRAK and D-TRAK. However, the difference is relatively small, which is maybe not too surprising given that the only difference between $\mathrm{GGN}_{\mathcal{D}}^{\mathrm{model}}$ and $\mathrm{GGN}_{\mathcal{D}}^{\mathrm{loss}}$ (E)K-FAC-expand is the distribution from which the labels are sampled (see Appendix C.2). In this sense, our $\mathrm{GGN}_{\mathcal{D}}^{\mathrm{loss}}$ (E)K-FAC expand approximation already fixes a potential problem (low-rankness) with the "empirical Fisher" $\mathrm{GGN}_{\mathcal{D}}^{\mathrm{loss}}$ used in TRAK and D-TRAK. As described in Appendix C.2 this is not the case for $\mathrm{GGN}_{\mathcal{D}}^{\mathrm{loss}}$ (E)K-FAC-reduce in Equation (34). As expected, the K-FAC-reduce approximation of $\mathrm{GGN}_{\mathcal{D}}^{\mathrm{loss}}$ performs worst among all the variants.[12] Secondly, we find that using the eigenvalue-corrected K-FAC (EK-FAC) variant, which should more closely approximate the respective GGN, improves results for all configurations. Thirdly, we find that K-FAC-expand noticeably outperforms K-FAC-reduce, which stands in contrast to some results in the second-order optimisation setting where the two are roughly on par with one another (Eschenhagen et al., 2023). This difference cannot be easily attributed since there are multiple differences between the two settings: we use a square loss instead of a cross entropy loss, a full dataset estimate, a different architecture, and evaluate the approximation in a different application. Notably, in some limited sense K-FAC-expand is better justified than K-FAC-reduce for our diffusion modelling setting, since it will be exact in the case of a (deep) linear network.

All our results seem to imply that a better Hessian approximation directly results in better downstream data attribution performance. However, we do not directly evaluate the approximation quality of the estimates.

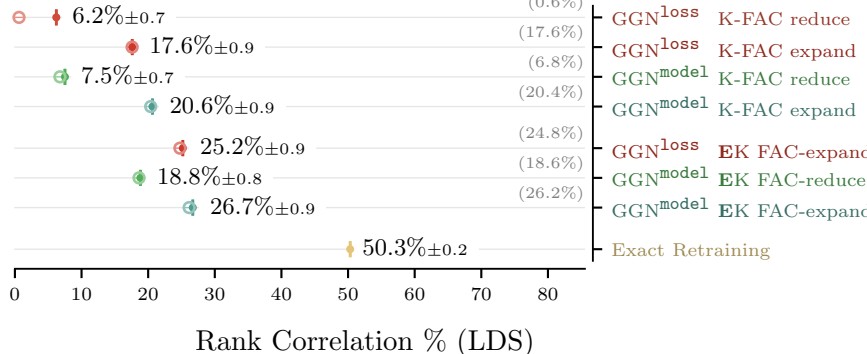

Figure 5: Ablation over the different Hessian approximations introduced in Section 3.1 and Appendix C.2. We ablate two versions of the GGN: the "MC" Fisher $\mathrm{GGN}^{\mathrm{model}}$ in Equation (9) and the "Empirical" Fisher $\mathrm{GGN}^{\mathrm{loss}}$ in Equation (11), as well as multiple settings for the K-FAC approximation: "expand" and "reduce", and whether we use the eigenvalue-corrected variant (EK-FAC) or not (K-FAC). Same as in Figure 2, we report the results for both the best damping value with ● and a default damping value of $10^{-8}$ with ○. The damping value ablation for the selection of these results is reported in Figure 11.

---

[12]Since the "reduce" variant for $\mathrm{GGN}_{\mathcal{D}}^{\mathrm{loss}}(\theta_l)$ in Equation (34) is not supported by `curvlinops` (Dangel et al., 2025), we only implemented this variant for regular K-FAC, and not for EK-FAC. However, we have no reason to expect the disappointing performance of the "reduce" variant with the $\mathrm{GGN}_{\mathcal{D}}^{\mathrm{loss}}$ would not persist for EK-FAC.

## D    RELATED WORK

**Data attribution in diffusion models** Data attribution in diffusion models has been tackled by extending TRAK (Park et al., 2023) in two previous works. Georgiev et al. (2023) apply TRAK to diffusion models, and argue that attributing the log-probability of sampling a latent from the sampling trajectory of an example at different diffusion timesteps will lead to attributions based on different semantic notions of similarity. In particular, directly attributing based on the diffusion loss of the generated sample leads to poor LDS scores. We also observe this is the case for TRAK without damping factor tuning in this work. Zheng et al. (2024), on the other hand, show that substituting a different (incorrect) measurement function and training loss into the TRAK approximation improves performance in terms of the LDS scores, and they recommend a particular setting based on empirical performance. Significant gains in performance in Zheng et al. (2024) can be observed from the tuning of a damping factor. Kwon et al. (2023) propose an approximation to the inverse of a Gauss-Newton matrix (specifically, the empirical Fisher) by heuristically interchanging the sum and the inverse operations; they apply influence functions with this Hessian approximation to LoRA-finetuned (Hu et al., 2021) diffusion models. However, they do not consider what GGN matrix to use for approximating the Hessian in diffusion models, resorting to the Empirical Fisher matrix, and they do not explore what measurement/loss function would be appropriate for this setting[13], which we do in this paper. In concurrent work, Lin et al. (2024) explore alternative measurement functions for data attribution in diffusion models with TRAK.

**Influence Functions** Influence functions were originally proposed as a method for data attribution in deep learning by Koh & Liang (2017a). Later, Koh et al. (2019) explored influence functions for investigating the effect of removing or adding groups of data points. Further extensions were proposed by Basu et al. (2020) — who explored utilising higher-order information — and Barshan et al. (2020), who aimed to improve influence ranking via re-normalisation. Initial works on influence functions (Koh & Liang, 2017a; Koh et al., 2019) relied on using iterative solvers to compute the required inverse-Hessian-vector products. Grosse et al. (2023) later explored using EK-FAC as an alternative solution to efficiently approximate calculations with the inverse Hessian.

For a broader overview of scalable training data attribution in the context of modern deep learning, see (Grosse et al., 2023, § 4).

## E    RUNTIME MEMORY AND COMPUTE

Influence functions can be implemented in different ways, caching different quantities at intermediate points, resulting in different trade-offs between memory and compute. A recommended implementation will also depend on whether one just wants to find most influential training examples for a selected set of query samples once, or whether one wants to implement influence functions in a system where new query samples to attribute come in periodically.

The procedure we follow in our implementation can roughly be summarised as informally depicted with pseudo-code in Algorithm 1.

For deployment, where new query samples might periodically come in, we might prefer to store compressed preconditioned training gradients instead. This is illustrated in Algorithm 2.

In principle, if we used an empirical Fisher approximation (like in Equation (10)) to approximate the GGN, we could further amortise the computation in the latter variant by caching training loss gradients during the K-FAC computation.

Note that, for applications like classification with a cross-entropy loss or auto-regressive language modelling Vaswani (2017), the gradients have a Kronecker structure, which means they could be stored much more efficiently Grosse et al. (2023). This is not the case for gradients of the diffusion loss in Section 2.1, since they require averaging multiple Monte-Carlo samples of the gradient.

We will primarily describe the complexities for the former variant (Algorithm 1), as that is the one we used for all experiments. The three sources of compute cost, which we will describe below, are: **1)** computing and inverting the Hessian, **2)** computing, pre-conditioning and compressing the query

---

[13]They state that they "used a negative log-likelihood of a generated image as a loss function", although they actually only use the diffusion training loss as a measurement.

---

**Algorithm 1** K-FAC Influence Computation (Single-Use)

---

1: **Input:** Training set $\{x_i\}_{i=1}^{N}$, query points $\{\hat{x}_j\}_{j=1}^{Q}$, number of Monte-Carlo samples $S \in \mathbb{N}$
2: **Output:** Influence scores $\{\text{score}_{ij} : i \in \{1, \ldots, N\}, j \in \{1, \ldots, Q\}\}$ for all pairings of training examples with query examples
3:    $H^{-1} \leftarrow \text{ComputeAndInvertKFAC}()$            ▷ Compute and store K-FAC inverse
4: **for** $j = 1$ to $Q$ **do**                ▷ Process query points
5:      $v_j \leftarrow \nabla_\theta m(\hat{x}_j; \theta)$          ▷ Compute gradient with $S$ samples
6:      $y_j \leftarrow H^{-1} v_j$               ▷ Precondition gradient
7:      Store compressed $y_j$
8: **end for**
9: **for** $i = 1$ to $N$ **do**               ▷ Process training points
10:     $v_i \leftarrow \nabla_\theta \ell(x_i; \theta)$          ▷ Compute gradient with $S$ samples
11:     **for** $j = 1$ to $Q$ **do**
12:        $\text{score}_{ij} \leftarrow y_j^\top v_i$         ▷ Compute influence score
13:     **end for**
14: **end for**

---

**Algorithm 2** K-FAC Influence Computation (Continual Deployment Setting)

---

1: **Input:** Training set $\{x_i\}_{i=1}^{N}$, samples $S$
2: **Output:** Cached preconditioned training gradients $\{y_i\}_{i=1}^{N}$ for efficient influence computation
3:    $H^{-1} \leftarrow \text{ComputeAndInvertKFAC}()$       ▷ Compute and store K-FAC inverse
4: **for** $i = 1$ to $N$ **do**               ▷ Preprocess training set
5:     $v_i \leftarrow \nabla_\theta \ell(x_i; \theta)$          ▷ Compute gradient with $S$ samples
6:     $y_i \leftarrow H^{-1} v_i$             ▷ Precondition gradient
7:     Store compressed $y_i$
8: **end for**
9: **procedure** COMPUTEINFLUENCE($\{\hat{x}_j\}_{j=1}^{Q}$)       ▷ Called when new queries arrive
10:     **for** $j = 1$ to $Q$ **do**
11:        $v_j \leftarrow \nabla_\theta \ell(\hat{x}_j; \theta)$        ▷ Compute query gradient
12:        **for** $i = 1$ to $N$ **do**
13:          $\text{score}_{ij} \leftarrow y_i^\top v_j$        ▷ Compute influence score
14:        **end for**
15:     **end for**
16:     **return** $\{\text{score}_{ij} : i \in \{1, \ldots, N\}, j \in \{1, \ldots, Q\}\}$
17: **end procedure**

---

gradients, and **3)** computing the training gradients and taking inner-products with the preconditioned query gradients.

### E.1 ASYMPTOTIC COMPLEXITY

Here, we will describe how runtime compute and memory scale with the number of query examples to attribute $Q$, the number of of training examples $N$, the number of Monte-Carlo samples $S$, for a standard feed-forward network with width $W$ and depth $L$[14]. These variables are summarised in Table 1. The number of parameters of the network $P$ is then $\mathcal{O}(W^2 L)$. [15]

Altogether, the runtime complexity of running K-FAC influence in this setting is $\mathcal{O}\left((N + Q)SW^2 L + NQW^2 L + W^3 L\right)$ and requires $\mathcal{O}(QW^2 L + NQ)$ storage. We break this down below.

---

[15] This can either be a multi-layer perceptron, or a convolutional neural network with $W$ denoting the channels. The feed-forward assumption is primarily chosen for illustrative purposes, but the analysis is straight-forward to extend to other architectures, and the asymptotic results do not differ for other common architectures.

| $Q$ | Number of query data points |
|---|---|
| $N$ | Number of training examples |
| $W$ | Maximum layer width |
| $L$ | Depth of the network |
| $S$ | Number of samples for Monte-Carlo evaluation of per-example loss or measurement gradients |

Table 1: Variables for scaling analysis.

### E.1.1    K-FAC AND K-FAC INVERSION

For each training example, and each sample, the additional computation of K-FAC over a simple forward-backward pass through the network (LeCun et al., 1988) comes from computing the outer products of post-activations, and gradients of loss with respect to the pre-activations. Overall, this adds a cost of $\mathcal{O}(W^2L)$ on top of the forward-backward pass, and so a single iteration has the same $\mathcal{O}(W^2L)$ cost scaling as a forward-backward pass. Hence, computing K-FAC for the entire training dataset with $S$ samples has cost $\mathcal{O}(NSW^2L)$.

Since K-FAC is a block-wise diagonal approximation, computing the inverse only requires computing the per-layer inverses. For a linear layer with input width $W_{\text{in}}$ and output width $W_{\text{out}}$, computing the inverse costs $\mathcal{O}(W_{\text{in}}^3 + W_{\text{out}}^3)$ due to the Kronecker-factored form of the K-FAC approximation. Similarly, storing K-FAC (or the inverse) requires storing matrices of sizes $W_{\text{in}} \times W_{\text{in}}$ and $W_{\text{out}} \times W_{\text{out}}$ for each linear layer.

Hence, computing K-FAC has a runtime complexity of $\mathcal{O}(NSLW^2)$. An additional $\mathcal{O}(LW^3)$ will be required for the inversion, which is negligible compared to the cost of computing K-FAC for larger datasets. The inverse K-FAC requires $\mathcal{O}(LW^2)$ storage. In practice, storing K-FAC (or inverse K-FAC) requires more memory than storing the network parameters, with the multiple depending on the ratios of layer widths across the network.

### E.1.2    PRECONDITIONED QUERY GRADIENTS COMPUTATION

Computing a single query gradient takes $\mathcal{O}(SW^2L)$ time, and preconditioning with K-FAC requires a further matrix-vector product costing $\mathcal{O}(W^3L)$. The cost of compressing the gradient will depend on the method, but, for quantisation (Appendix F), it's negligible compared to the other terms. Hence, computing all $Q$ query gradients costs $\mathcal{O}(QSW^2L + QW^3L)$.

Storing the $Q$ preconditioned gradients requires $\mathcal{O}(QW^2L)$ storage (although, in principle, this could be more efficient depending on the compression method chosen and how it scales with the network size while maintaining precision).

### E.1.3    TRAINING GRADIENTS AND SCORES COMPUTATION

Again, computing a single training gradient takes $\mathcal{O}(SW^2L)$, and an inner product with all the preconditioned query gradients takes an additional $\mathcal{O}(QW^2L)$. Hence, this part requires $\mathcal{O}(NSW^2L + NQW^2L)$ operations.

Storage-wise, storing the final "scores" (the preconditioned inner products between the training and query gradients) requires a further $\mathcal{O}(NQ)$ memory, but this is typically small ($4NQ$ bytes for `float32` precision).

### E.1.4    COMPARISON WITH TRAK

The complexity of TRAK (Park et al., 2023) additionally depends on the choice of the projection dimension $R$. The computational cost of running TRAK is $\mathcal{O}((N + Q)SW^2L + NQR + R^3)$. Similarly, the memory cost of the implementation by Park et al. (2023) is $\mathcal{O}((N + Q)R + R^2)$.

Note that, it is unclear how $R$ should scale with the neural network size $W^2L$. Random projections do allow for constant scaling with vector size while maintaining approximation quality in some settings (Johnson et al., 1986). To the best of our knowledge, it has not been shown, either empirically or

theoretically, what the expected scaling of $R$ with network size might be in the context of influence function preconditioned inner products (Equation (6)). In the worst case, the projection dimension $R$ might be required to scale proportionally to the network size to maintain a desired level of accuracy.

## E.2 RUNTIME COMPLEXITY

We also report the runtimes of computing TRAK and K-FAC influence scores for the experiments reported in this paper. We discuss what additional memory requirements one might expect when running these methods. All experiments were ran on a single NVIDIA A100 GPU.

The runtime and memory is reported for computing influence for 200 query data points. As discussed at the beginning of Appendix E, K-FAC computation and inversion costs are constant with respect to the number of query data points, and computing the training gradients can be amortised in a sensible deployment-geared implementation at the added memory cost of storing the (compressed) training gradients.

### E.2.1 RUNTIME RESULTS

Tables 2 and 3 report the runtimes on a single NVIDIA A100 GPU of the most time-consuming parts of the influence function computation procedure.

| Dataset (size) | Dataset size | # network param. (millions) | # MC samples | K-FAC computation | Query gradients | Training gradients |
|---|---|---|---|---|---|---|
| CIFAR-2 | 5000 | 38.3 | 250 | 03:30:32 | 01:12 | 00:34:42 |
| CIFAR-10 | 50000 | 38.3 | 250 | 35:01:33 | 01:12 | 05:43:14 |
| ArtBench | 50000 | 37.4 +83.6* | 125 | 32:48:08 | 04:18 | 18:56:57 |

Table 2: Runtime for K-FAC influence score computation across datasets. "*" indicates parameters of a pre-trained part of the model (e.g. VAE for Latent Diffusion Models).

| Dataset (size) | Dataset size | # network param. (millions) | # MC samples | Query gradients | Training gradients | Hessian inversion | Computing scores |
|---|---|---|---|---|---|---|---|
| CIFAR-2 | 5000 | 38.3 | 250 | 3:38 | 01:32:10 | 00:11 | 3:43 |
| CIFAR-10 | 50000 | 38.3 | 250 | 3:44 | 15:15:04 | 00:57 | 3:54 |
| ArtBench | 50000 | 37.4 +83.6* | 125 | 5:46 | 23:58:44 | 00:28 | 6:24 |

Table 3: Runtime for TRAK score computation across datasets. "*" indicates parameters of a pre-trained part of the model (e.g. VAE for Latent Diffusion Models).

### E.2.2 MEMORY USAGE

Tables 4 and 5 report the expected memory overheads due to having to manifest and store large matrices or collections of vectors in the influence function implementations of K-FAC Influence and TRAK.

| Dataset | # network param. (millions) | Inverse K-FAC (GB) | Cached query gradients (GB) |
|---|---|---|---|
| CIFAR-2 | 38.3 | 1.57 | 7.66 |
| CIFAR-10 | 38.3 | 1.57 | 7.66 |
| ArtBench | 37.4 +83.6* | 1.57 | 7.47 |

Table 4: Memory usage linked to K-FAC Influence. "*" indicates parameters of a pre-trained part of the model (e.g. VAE for Latent Diffusion Models).

| Dataset | # network param. (millions) | Projection dimension | Projected train gradients (GB) | Projected query gradients (GB) | Projected Hessian inverse (GB) |
|---|---|---|---|---|---|
| CIFAR-2 | 38.3 | 32768 | 0.66 | 0.027 | 4.29 |
| CIFAR-10 | 38.3 | 32768 | 6.55 | 0.027 | 4.29 |
| ArtBench | 37.4 +83.6* | 32768 | 6.55 | 0.027 | 4.29 |

Table 5: Memory usage linked to TRAK. "*" indicates parameters of a pre-trained part of the model (e.g. VAE for Latent Diffusion Models).

## F  GRADIENT COMPRESSION ABLATION

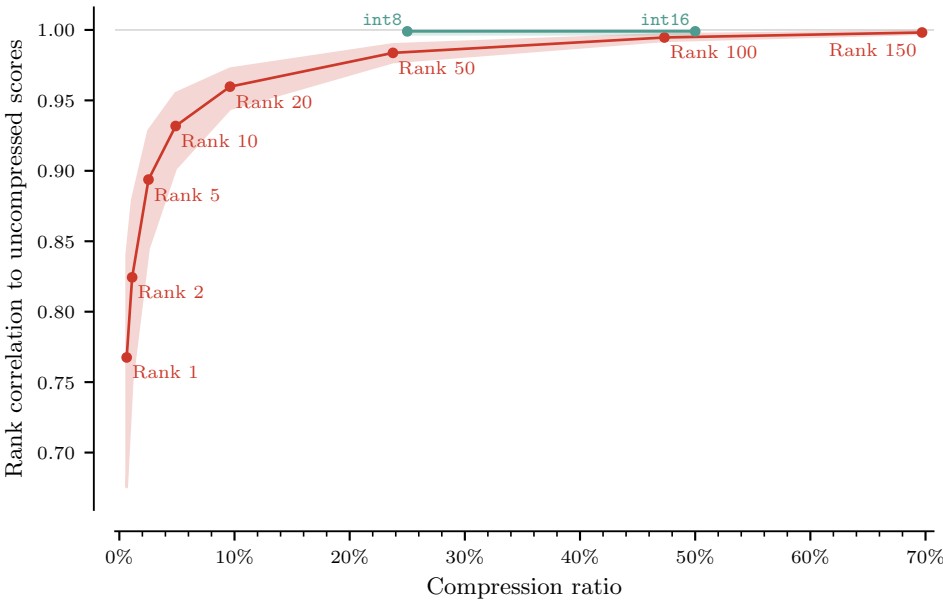

Figure 6: Comparison of gradient compression methods for the influence function approximation.

In Figure 6, we ablate different compression methods by computing the per-training-datapoint influence scores with both **a)** compressed query (measurement) gradients, and **b)** the uncompressed gradients, and comparing the Pearson and rank correlations between the scores computed with (a) and (b). We hope to see a correlation of close to $100\%$, in which case the results for our method would be unaffected by compression. We find that using $8$-bit quantisation for compression results in almost no change to the ordering over training datapoints. This is in contrast to the SVD compression scheme used in Grosse et al. (2023). This is likely because the per-example gradients naturally have a low-rank (Kronecker) structure in the classification, regression, or autoregressive language modelling settings, such as the one considered by Grosse et al. (2023). On the other hand, the diffusion training loss and other measurement functions considered in this work do not have this low-rank structure. This is because computing them requires multiple forward passes; for example, for the diffusion training loss, we need to average the mean square error loss in Equation (2) over multiple noise samples $\epsilon^{(t)}$ and multiple diffusion timesteps. We use $8$ bit quantisation with query gradient batching (Grosse et al., 2023) for all K-FAC experiments throughout this work.

## G  DAMPING LDS ABLATIONS

We report an ablation over the LDS scores with the GGN approximated with different damping factors for TRAK/D-TRAK and K-FAC influence in Figures 7 to 12. The reported damping factors for TRAK are normalised by the dataset size so that they correspond to the equivalent damping factors for our method when viewing TRAK as an alternative approximation to the GGN (see Section 3.1).

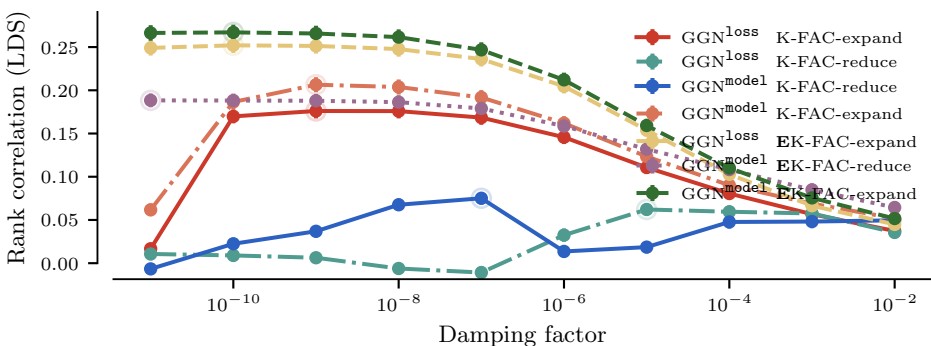

Figure 7: Effect of damping on the LDS scores for **K-FAC influence** on `CIFAR-2`. 250 samples were used for Monte Carlo estiamtion of all quantities (GGN and the training loss/measurement gradients). `Target`/`Measure` indicate what measurement was used for the ground-truth and in the approximation respectively.

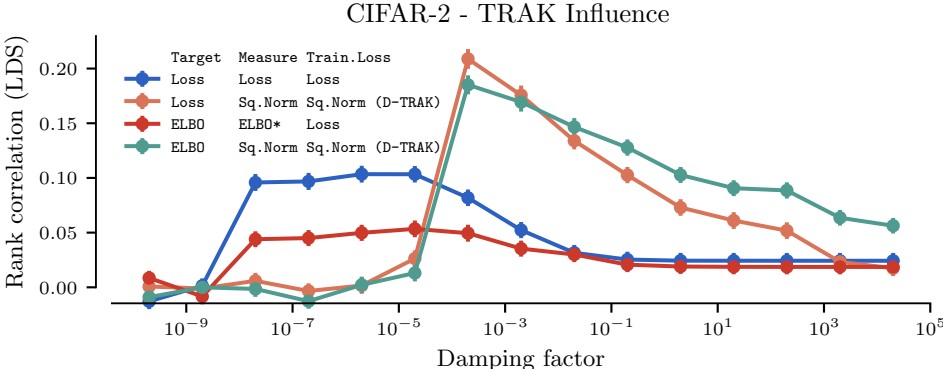

Figure 8: Effect of damping on the LDS scores for TRAK (random projection) based influence on `CIFAR-2`. 250 samples were used for Monte Carlo estiamtion of all quantities (GGN and the training loss/measurement gradients). In the legend: `Target` indicates what measurement we're trying to predict the change in after retraining, `Measure` indicates what measurement function was substituted into the influence function approximation, and `Train.Loss` indicates what function was substituted for the training loss in the computation of the GGN and gradient of the training loss in the influence function approximation.

## H    EMPIRICAL ABLATIONS FOR CHALLENGES TO USE OF INFLUENCE FUNCTIONS FOR DIFFUSION MODELS

In this section, we describe the results for the observations discussed in Section 4.1.

**Observation 1** is based on Figures 13 and 14. Figure 13 shows the LDS scores on `CIFAR-2` when attributing per-timestep diffusion losses $\ell_t$ (see Equation (2)) using influence functions, whilst varying what (possibly wrong) per-timestep diffusion loss $\ell_{t'}$ is used as a measurement function in the influence function approximation (Equation (6)). Figure 14 is a counter-equivalent to Figure 18 where instead of using influence functions to approximate the change in measurement, we actually retrain a model on the randomly subsampled subset of data and compute the measurement.

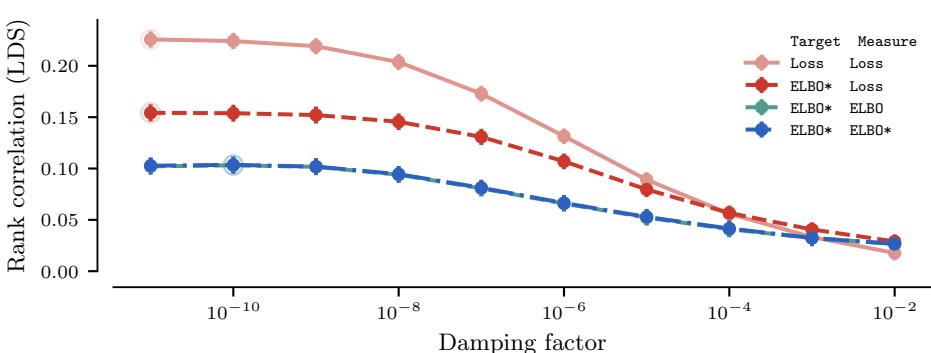

Figure 9: Effect of damping on the LDS scores for K-FAC based influence on `CIFAR-10`. 250 samples were used for computing the EK-FAC GGN approximation (125 for the eigenbasis computations, 125 for the eigenvalue computations), and 250 for computing a Monte Carlo estimate of the training loss/measurement gradients. `Target`/`Measure` indicate what measurement was used for the ground-truth and in the approximation respectively.

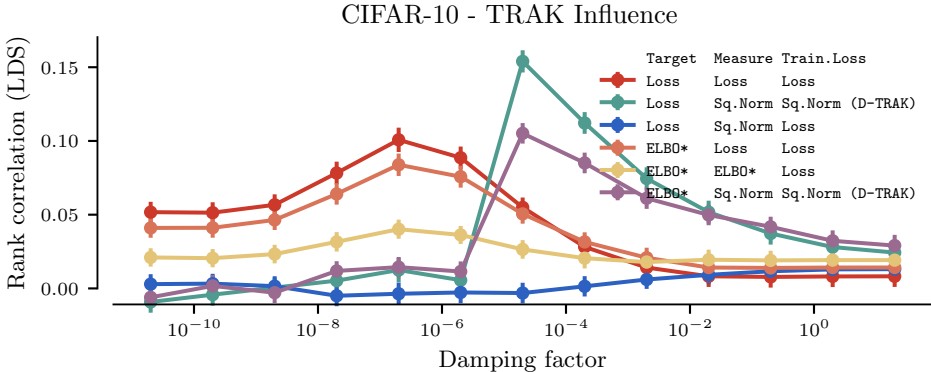

Figure 10: Effect of damping on the LDS scores for TRAK (random projection) based influence on `CIFAR-10`. 250 samples were used for Monte Carlo estiamtion of all quantities (GGN and the training loss/measurement gradients). In the legend: `Target` indicates what measurement we're trying to predict the change in after retraining, `Measure` indicates what measurement function was substituted into the influence function approximation, and `Train.Loss` indicates what function was substituted for the training loss in the computation of the GGN and gradient of the training loss in the influence function approximation.

A natural question to ask with regards to Observation 1 is: does this effect go away in settings where the influence function approximation should more exact? Note that, bar the non-convexity of the training loss function $\mathcal{L}_{\mathcal{D}}$, the influence function approximation in Equation (6) is a linearisation of the actual change in the measurement for the optimum of the training loss functions with some examples down-weighted by $\varepsilon$ around $\varepsilon = 0$. Hence, we might expect the approximation to be more exact when instead of fully removing some data points from the dataset (setting $\varepsilon = 1/N$), we instead down-weight their contribution to the training loss by a smaller non-zero factor. To investigate whether this is the case, we repeat the LDS analysis in Figures 13 and 14, but with $\varepsilon = 1/2N$; in other words, the training loss terms corresponding to the "removed" examples are simply down-weighted by a factor of $1/2$ in the retrained models. The results are shown in Figures 15 and 16. Perhaps somewhat surprisingly, a contrasting effect can be observed, where using per-timestep diffusion losses for larger times yields a higher absolute rank correlation, but with the opposing sign. The negative

CIFAR-2 - K-FAC Influence Ablation

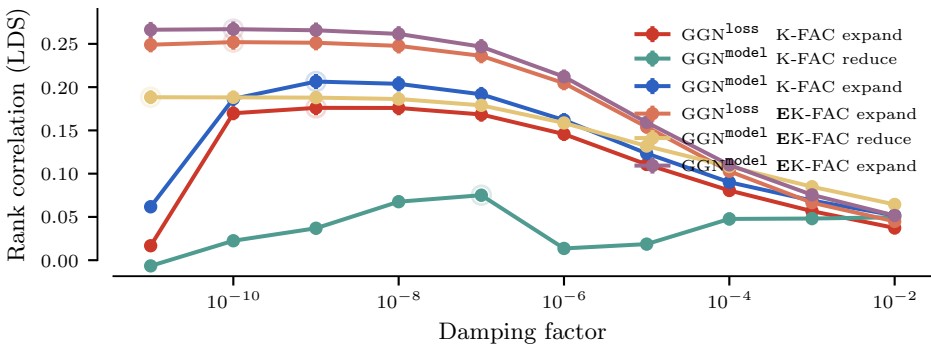

Figure 11: Effect of damping on the LDS scores for various GGN K-FAC approximations on CIFAR-2. 250 samples were used for Monte Carlo estiamtion of all quantities (GGN and the training loss/measurement gradients); for eigenvalue-corrected K-FAC, the 250 samples are split so that 125 samples are used for the computation of the eigen-basis, and 125 samples are used for the computation of the eigen-values, in order to roughly match the runtime between K-FAC and EK-FAC.

CIFAR-2 (deq.) - K-FAC Influence

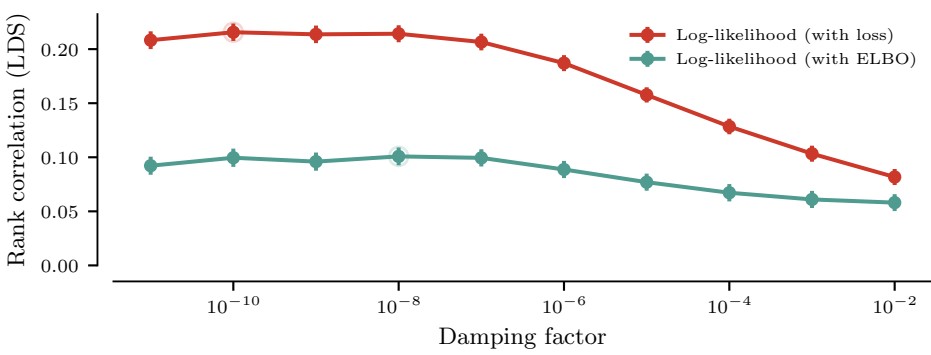

Figure 12: Effect of damping on the LDS scores for K-FAC influence for approximating the marginal log-probability measurement on dequantised CIFAR-2.

correlation between measurement $\ell_t, \ell_{t'}$ for $t \neq t'$ can also be observed for the true measurements in the retrained models in Figure 16. We also observe that in this setting, influence functions fail completely to predict changes in $\ell_t$ with the correct measurement function for $t \leq 200$.

**Observation 2** Figure 17 shows the changes in losses after retraining the model on half the data removed against the predicted changes in losses using K-FAC Influence for two datasets: CIFAR-2 and CIFAR-10. In both cases, for a vast majority of retrained models, the loss measurement on a sample increases after retraining. On the other hand, the influence functions predict roughly evenly that the loss will increase and decrease. This trend is amplified if we instead look at influence predicted for per-timestep diffusion losses $\ell_t$ (Equation (2)) for earlier timesteps $t$, which can be seen in Figure 18. On CIFAR-2, actual changes in $\ell_1, \ell_50, \ell_100$ measurements are actually *always* positive, which the influence functions approximation completely misses. For all plots, K-FAC Influence was ran with a damping factor of $10^{-8}$ and 250 samples for all gradient computations.

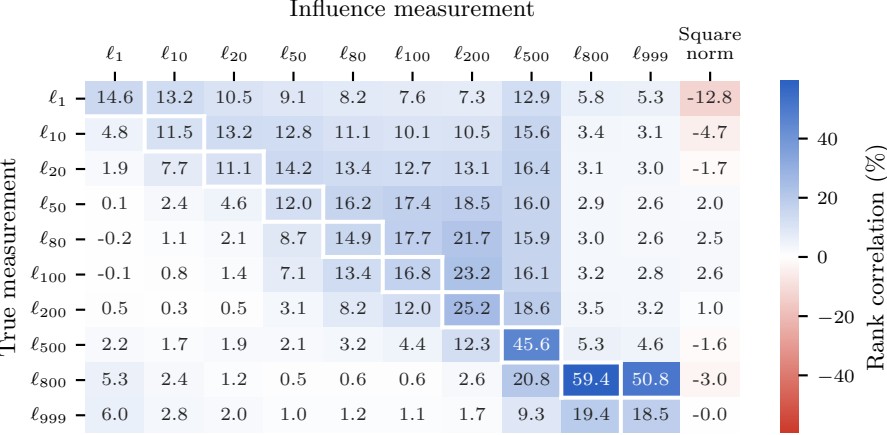

Figure 13: Rank correlation (LDS scores) between influence function estimates with different measurement functions and different true measurements `CIFAR-2`. The plot shows how well different per-timestep diffusion losses $\ell_t$ work as measurement functions in the influence function approximation, when trying to approximate changes in the actual measurements when retraining a model.

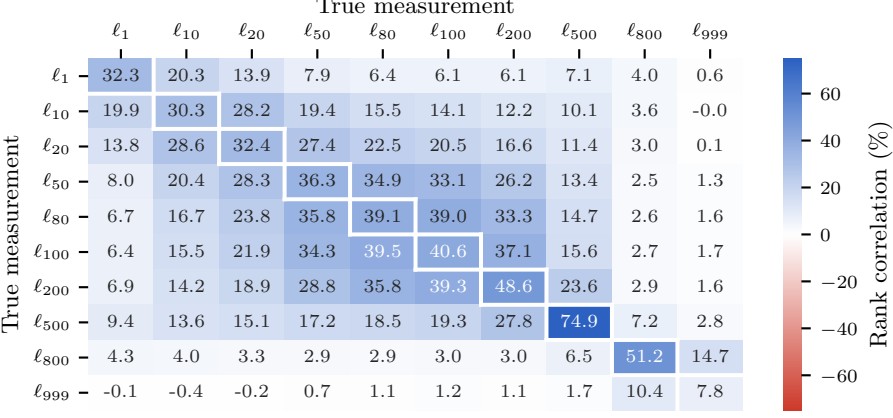

Figure 14: Rank correlation between true measurements for losses at different diffusion timesteps on `CIFAR-2`.

Figures 17 and 18 also shows that influence functions tend to overestimate the *magnitude* of the change in measurement after removing the training data points. This is in contrast to the observation in (Koh et al., 2019) in the supervised setting, where they found that influence functions tend to *underestimate* the magnitude of the change in the measurement. There are many plausible reasons for this, ranging from the choice of the Hessian approximation ((Koh et al., 2019) compute exact inverse-Hessian-vector products, whereas we approximate the GGN), to the possible "stability" of the learned distribution in diffusion models even when different subsets of data are used for training (Observation 3 and (Kadkhodaie et al., 2024)).

**Observation 3** Lastly, the observation that marginal log-probability remains essentially constant for models trained on different subsets of data is based on Figures 19a to 19c.

In that figure, we reproduce a similar behaviour to that observed by Kadkhodaie et al. (2024) – the samples generated from the models are all different for models trained on subsets below a certain size, and they are nearly identical for models trained on subsets past a critical size. This "collapse" also happens for the models' log-likelihoods (marginal log-probability density), as well as the other

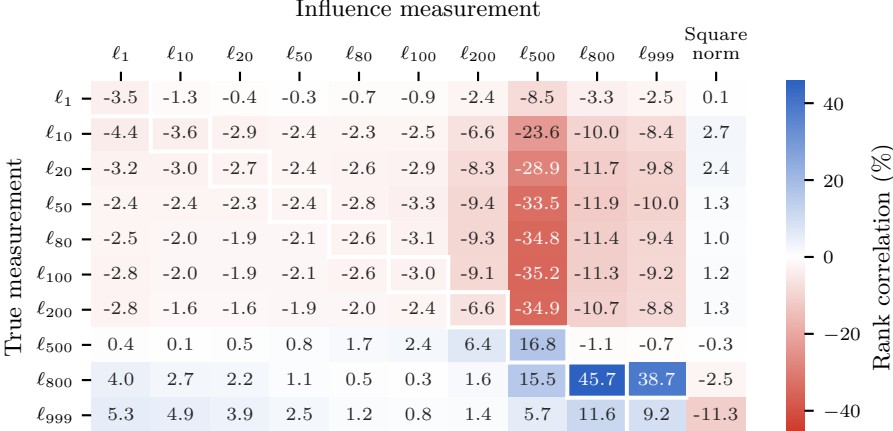

Figure 15: Rank correlation (LDS scores) between influence function estimates with different measurement functions and different true measurements `CIFAR-2`, but with the retrained models trained on the full dataset with a random subset of examples having a **down-weighted contribution to a training loss by a factor of** $\times 0.5$.

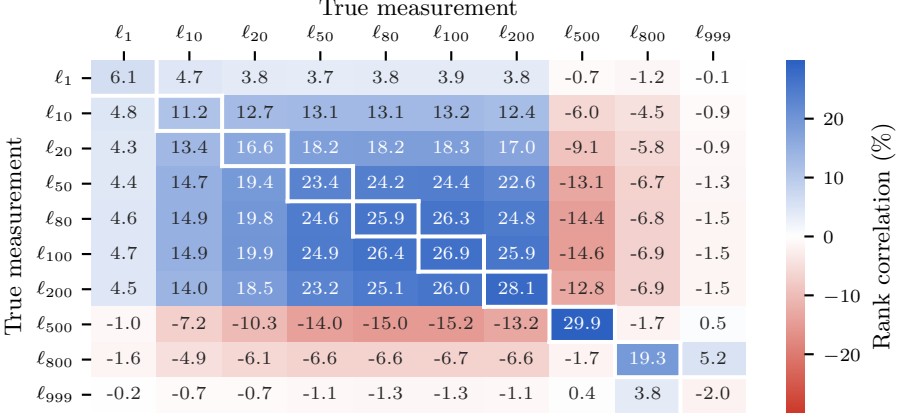

Figure 16: Rank correlation between true measurements for losses at different diffusion timesteps on `CIFAR-2`, but with the retrained models trained on the full dataset with a random subset of examples having a **down-weighted contribution to a training loss by a factor of** $\times 0.5$.

measurements considered in this work (ELBO and training loss). Past a certain subset size threshold, ELBO, loss and the log-likelihood of models retrained on different subsets become identical. Before that subset size threshold, there is some difference among the models. The threshold seems to be smaller for the ELBO compared to the loss measurement. We have computed these quantities with models trained on uniformly dequantised `CIFAR-10` Song et al. (2021b) so that we can meaningfully compute the marginal log-probability density, but we have also verified this observation for the ELBO and the training loss measurements on regular `CIFAR-10`. Each measurement was computed with a Monte-Carlo estimate using 5000 samples, except for log-likelihood, where the measurement was averaged over 5 uniformly dequantised samples.

## I  LDS RESULTS FOR THE MARGINAL LOG-PROBABILITY MEASUREMENT

The results for the "marginal log-probability" measurement are shown in Figure 21. This measurement uses the interpretation of the diffusion model as a score estimator Song et al. (2021b), and computes the marginal log-probability density assuming a normalising flow reformulation of the model Song

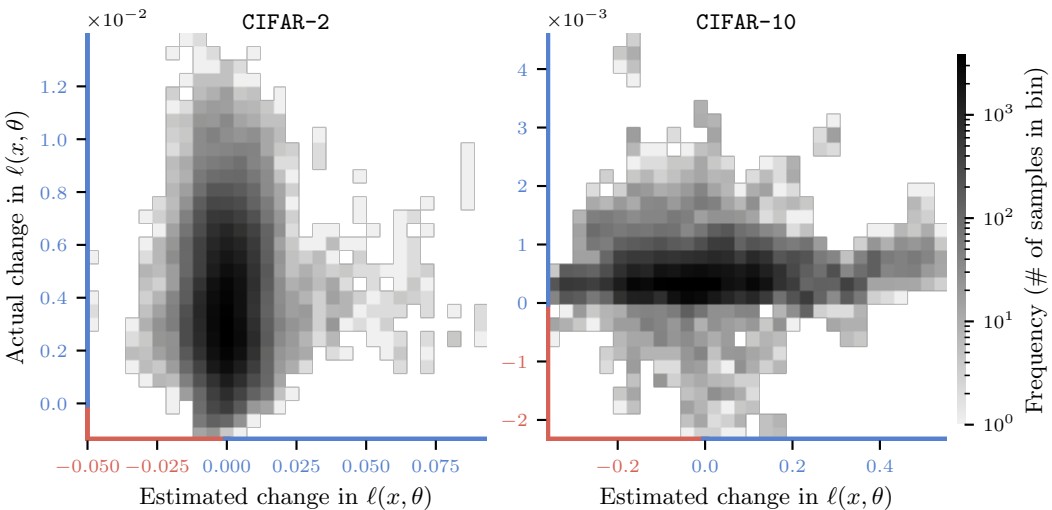

Figure 17: Change in diffusion loss $\ell$ in Section 2.1 when retraining with random subsets of 50% of the training data removed, as predicted by K-FAC influence ($x$-axis), against the actual change in the measurement ($y$-axis). Results are plotted for measurements $\ell(x, \theta)$ for 50 samples $x$ generated from the diffusion model trained on all of the data. The scatter color indicates the sample $x$ for which the change in measurement is plotted. The figure shows that influence functions tend to overestimate how often the loss will decrease when some training samples are removed; in reality, it happens quite rarely.

et al. (2021b). Note that this will in general be different from the marginal log-density under the stochastic differential equation (SDE) model that was used for sampling Song et al. (2021a). For these measurements, we trained models on the *dequantised* CIFAR-2 dataset as described in Song et al. (2021b). This was done so that the empirical data distribution on which we train the models has a density. Note that this is different from the setting targeted by Ho et al. (2020) in which the data is assumed to live in a discrete space.

## J    EXPERIMENTAL DETAILS

In this section, we describe the implementation details for the methods and baselines, as well as the evaluations reported in Section 4.

### J.1    DATASETS

We focus on the following datasets in this paper:

**CIFAR-10** CIFAR-10 is a dataset of small RGB images of size $32 \times 32$ Krizhevsky (2009). We use 50000 images (the train split) for training.

**CIFAR-2** For CIFAR-2, we follow Zheng et al. (2024) and create a subset of CIFAR-10 with 5000 examples of images only corresponding to classes car and horse. 2500 examples of class car and 2500 examples of class horse are randomly subsampled without replacement from among all CIFAR-10 images of that class.

**ArtBench-10** The ArtBench-10 dataset (Liao et al., 2022) is a dataset of 60000 artworks from 10 artistic styles. The RGB images of the artworks are standardised to a $256 \times 256$ resolution. We use the full original train-split (50000 examples) from the original paper (Liao et al., 2022) for our experiments.

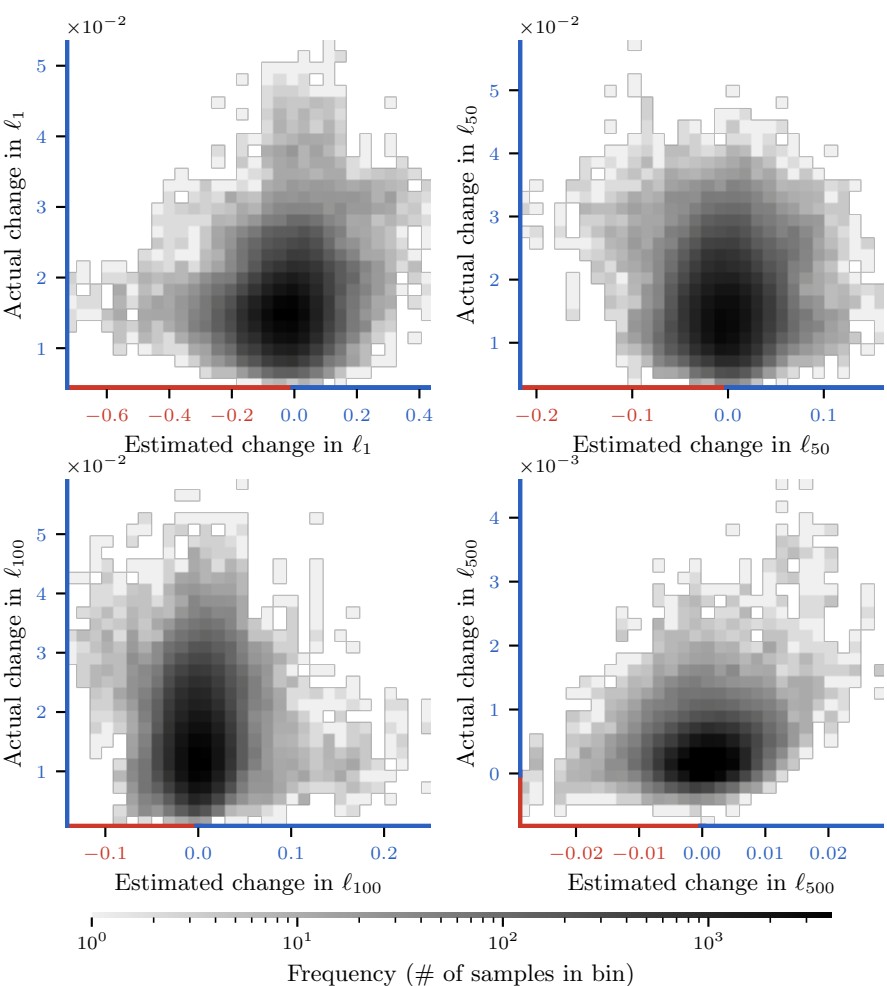

Figure 18: Change in per-diffusion-timestep losses $\ell_t$ when retraining with random subsets of 50% of the training data removed, as predicted by K-FAC influence ($x$-axis), against the actual change in the measurement ($y$-axis). Results are plotted for the CIFAR-2 dataset, for measurements $\ell_t(x, \theta)$ for 50 samples $x$ generated from the diffusion model trained on all of the data. The scatter color indicates the sample $x$ for which the change in measurement is plotted. The figure shows that: **1)** influence functions predict that the losses $\ell_t$ will increase or decrease roughly equally frequently when some samples are removed, but, in reality, the losses almost always increase; **2)** for sufficiently large time-steps ($\ell_5 00$), this pattern seems to subside. Losses $\ell_t$ in the $200 - 500$ range seem to work well for predicting changes in other losses Figure 13.

## J.2 Models

**CIFAR** For all CIFAR datasets, we train a regular Denoising Diffusion Probabilistic Model using a standard U-Net architecture as described for CIFAR-10 in Ho et al. (2020). This U-Net architecture contains both convolutional and attention layers. We use the same noise schedule as described for the CIFAR dataset in Ho et al. (2020).

**ArtBench** For the ArtBench-10 experiments, we use a Latent Diffusion Model (Rombach et al., 2022) with the diffusion backbone trained from scratch. The architecture for the diffusion in the latent space is based on a U-Net with transformer layers, and is fully described in the codebase at https://github.com/BrunoKM/diffusion-influence. For the latent-space encoder-decoder, we use the pretrained autoencoder from Stable Diffusion version 2 (Rombach et al., 2022), and fix its parameters throughout training.

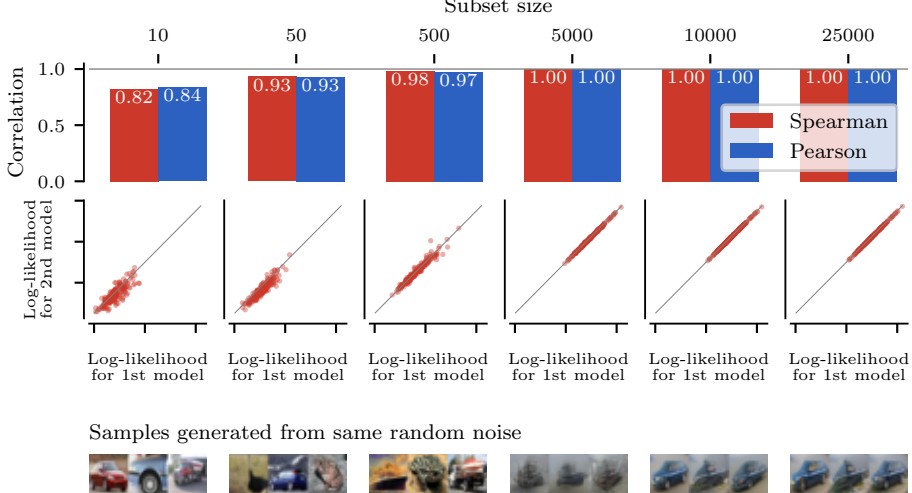

(a) Correlations in the **marginal log-probability density** for models trained on independent training sets of different sizes.

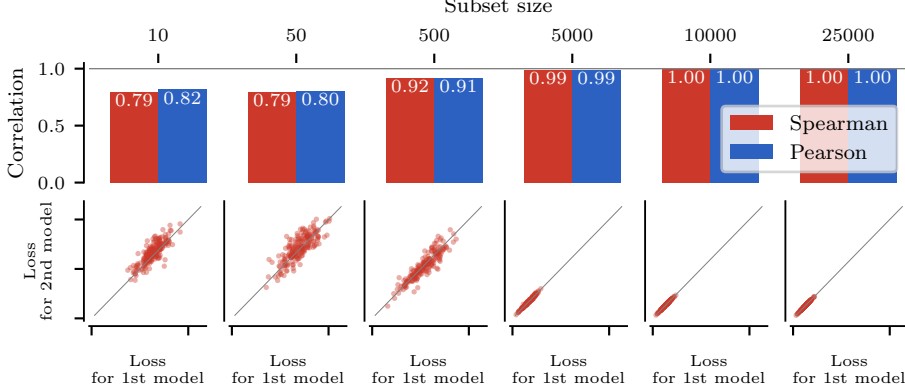

(b) Correlations in the **loss** measurement for models trained on independent training sets of different sizes.

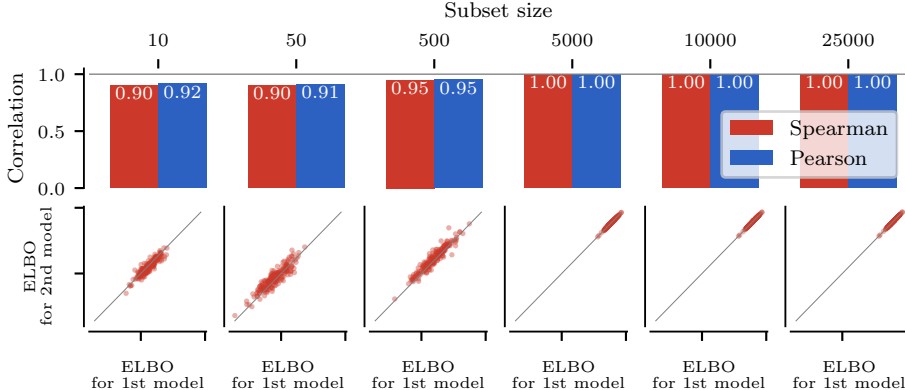

(c) Correlations in the **ELBO** measurement for models trained on independent training sets of different sizes.

Figure 19: Marginal log-probability density, the generated samples, the ELBO and the loss measurements eventually collapse to being near-identical for diffusion models trained on independent training sets of a sufficient size.

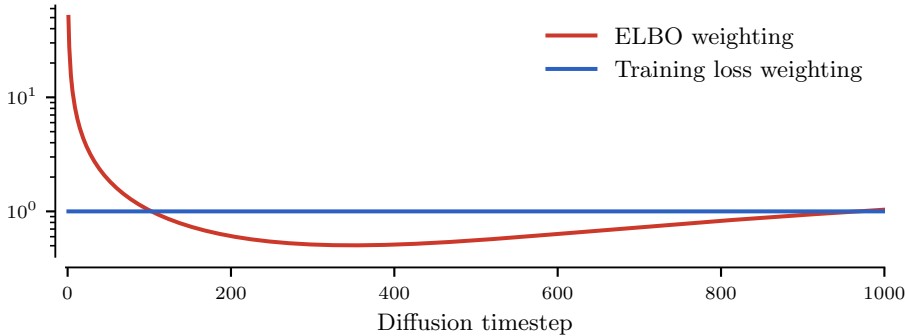

Figure 20: The diffusion loss and diffusion ELBO as formulated in (Ho et al., 2020) (ignoring the reconstruction term that accounts for the quantisation of images back to pixel space) are equal up to the weighting of the individual per-diffusion-timestep loss terms and a constant independent of the parameters. This plot illustrates the relatives difference in the weighting for per-diffusion-timestep losses applied in the ELBO vs. in the training loss.

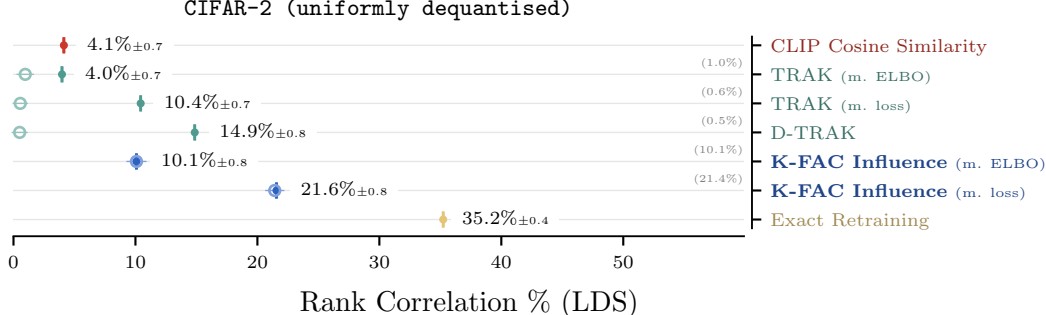

Figure 21: Linear Data-modelling Score (LDS) for the **marginal log-probability density** measurement. The plot follows the same format as that of Figures 2a and 2b, with the exception that the colours instead only represent method family groupings. Overall, the method and the proxies proposed in this paper seem to work well on estimating the changes in the marginal log-probability.

## J.3 SAMPLING

For the DDPM models (`CIFAR`), we follow the standard DDPM sampling procedure with a full 1000 timesteps to create the generated samples as described by Ho et al. (2020). DDPM sampling usually gives better samples (in terms of visual fidelity) than Denoising Diffusion Implicit Models (DDIM) sampling Song et al. (2022) when a large number of sampling steps is used. As described in Section 2.1, when parameterising the conditionals $p_\theta(x^{(t-1)}|x^{(t)})$ with neural networks as $\mathcal{N}\big(x^{(t-1)}|\mu_{t-1|t,0}\big(x^{(t)}, \epsilon_\theta^t(x^{(t)})\big), \sigma_t^2 I\big)$ we have a choice in how to set the variance hyperparameters $\{\sigma_t^2\}_{t=1}^T$. The $\sigma_t^2$ hyperparameters do not appear in the training loss; however, they do make a difference when sampling. We use the "small" variance variant from Ho et al. (2020, §3.2), i.e. we set:

$$\sigma_t^2 = \frac{1 - \prod_{t'=1}^{t-1} \lambda_{t'}}{1 - \prod_{t'=1}^{t} \lambda_{t'}} (1 - \lambda_t)$$

For `ArtBench-10` experiments, we follow (Rombach et al., 2022) using the full 1000 timesteps for sampling in the latent space before decoding the sample.

### J.4 DETAILS ON DATA ATTRIBUTION METHODS

**TRAK** For TRAK baselines, we adapt the implementation of Park et al. (2023); Georgiev et al. (2023) to the diffusion modelling setting. When running TRAK, there are several settings the authors recommend to consider: **1)** the projection dimension $d_{\texttt{proj}}$ for the random projections, **2)** the damping factor $\lambda$, and **3)** the numerical precision used for storing the projected gradients. For **(1)**, we use a relatively large projection dimension of 32768 as done in most experiments in Zheng et al. (2024). We found that the projection dimension affected the best obtainable results significantly, and so we couldn't get away with a smaller one. We also found that using the default `float16` precision in the TRAK codebase for **(3)** results in significantly degraded results (see Figure 22, and so we recommend using `float32` precision for these methods for diffusion models. In all experiments, we use `float32` throughout. For the damping factor, we report the sweeps over LDS scores in Figures 8 and 10, and use the best result in each benchmark, as these methods fail drastically if the damping factor is too small. The damping factor reported in the plots is normalised by the dataset size $N$, to match the definition of the GGN, and to make it comparable with the damping reported for other influence functions methods introduced in this paper. For non-LDS experiments, we use the best damping value from the corresponding LDS benchmark.

**CLIP cosine similarity** One of the data attribution baselines used for the LDS experiments is CLIP cosine similarity (Radford et al., 2021). For this baseline, we compute the CLIP embeddings (Radford et al., 2021) of the generated sample and training datapoints, and consider the cosine similarity between the two as the "influence" of that training datapoint on that particular target sample. See (Park et al., 2023) for details of how this influence is aggregated for the LDS benchmark. Of course, this computation does not in any way depend on the diffusion model or the measurement function used, so it is a pretty naïve method for estimating influence.

**(E)K-FAC** We use `curvlinops` (Dangel et al., 2025) package for our implementation of (E)K-FAC for diffusion models. Except where explicitly mentioned otherwise, we use the K-FAC (or EK-FAC) expand variant throughout. We compute (E)K-FAC for PyTorch `nn.Conv2d` and `nn.Linear` modules (including all linear maps in attention), ignoring the parameters in the normalisation layers.

**Compression** for all K-FAC influence functions results, we use `int8` quantisation for the query gradients.

**Monte Carlo computation of gradients and the GGN for influence functions** Computing the per-example training loss $\ell(\theta, x_n)$ in Section 2.1, the gradients of which are necessary for computing the influence function approximation (Equation (6)), includes multiple nested expectations over diffusion timestep $\tilde{t}$ and noise added to the data $\epsilon^{(t)}$. This is also the case for the $\text{GGN}_{\mathcal{D}}^{\texttt{model}}$ in Equation (9) and for the gradients $\nabla_\theta \ell(\theta, x_n)$ in the computation of $\text{GGN}_{\mathcal{D}}^{\texttt{loss}}$ in Equation (11), as well as for the computation of the measurement functions. Unless specified otherwise, we use the same number of samples for a Monte Carlo estimation of the expectations for all quantities considered. For example, if we use $K$ samples, that means that for the computation of the gradient of the per-example-loss $\nabla_\theta \ell(\theta, x_n)$ we'll sample tuples of $(\tilde{t}, \epsilon^{(\tilde{t})}, x^{(\tilde{t})})$ independently $K$ times to form a Monte Carlo estimate. For $\text{GGN}_{\mathcal{D}}^{\texttt{model}}$, we explicitly iterate over all training data points, and draw $K$ samples of $\left(\tilde{t}, \epsilon^{(\tilde{t})}, x_n^{(\tilde{t})}\right)$ for each datapoint. For $\text{GGN}_{\mathcal{D}}^{\texttt{loss}}$, we explicitly iterate over all training data points, and draw $K$ samples of $\left(\tilde{t}, \epsilon^{(\tilde{t})}, x_n^{(\tilde{t})}\right)$ to compute the gradients $\nabla_\theta \ell(\theta, x_n)$ before taking an outer product. Note that, for $\text{GGN}_{\mathcal{D}}^{\texttt{loss}}$, because we're averaging over the samples before taking the outer product of the gradients, the estimator of the GGN is no longer unbiased. Similarly, $K$ samples are also used for computing the gradients of the measurement function.

For all `CIFAR` experiments, we use 250 samples throughout for all methods (including all gradient and GGN computations for K-FAC Influence, TRAK, D-TRAK), unless explicitly indicated in the caption otherwise.

### J.5 DAMPING

For all influence function-like methods (including TRAK and D-TRAK), we use damping to improve the numerical stability of the Hessian inversion. Namely, for any method that computes the inverse of the approximation to the Hessian $H \approx \nabla_\theta^2 \mathcal{L}_\mathcal{D} = \nabla_\theta^2 1/N \sum \ell(\theta, x_n)$, we add a damping factor $\lambda$ to

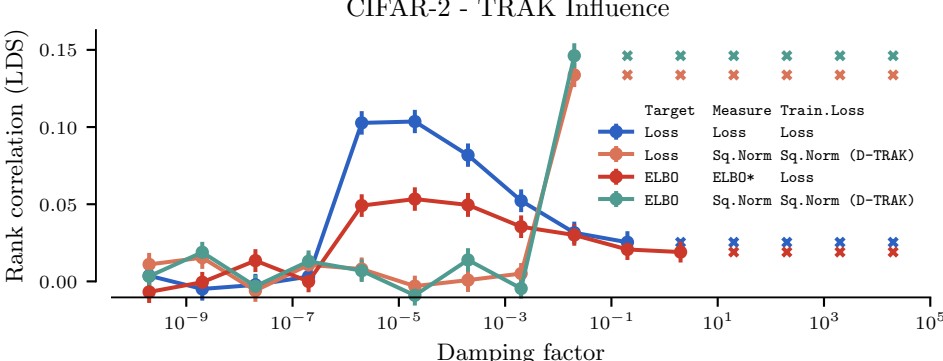

Figure 22: LDS scores on for TRAK (random projection) based influence on `CIFAR-2` when using half-precision (**`float16`**) for influence computations. Compare with Figure 8. `NaN` results are indicated with ✕.

the diagonal before inversion:

$$(H + \lambda I)^{-1},$$

where $I$ is a $d_{\texttt{param}} \times d_{\texttt{param}}$ identity matrix. This is particularly important for methods where the Hessian approximation is at a high risk of being low-rank (for example, when using the empirical GGN in Equation (11), which is the default setting for TRAK and D-TRAK). For TRAK/D-TRAK, the approximate Hessian inverse is computed in a smaller projected space, and so we add $\lambda$ to the diagonal directly in that projected space, as done in Zheng et al. (2024)). In other words, if $P \in \mathbb{R}^{d_{\texttt{proj}} \times d_{\texttt{param}}}$ is the projection matrix (see (Park et al., 2023) for details), then damped Hessian-inverse preconditioned vector inner products between two vectors $v_1, v_2 \in \mathbb{R}^{d_{\texttt{param}}}$ (e.g. the gradients in Equation (6)) would be computed as:

$$(Pv_1)^{\mathsf{T}} \left(H + \lambda I\right)^{-1} Pv_,.$$

where $H \approx P \nabla_\theta^2 \mathcal{L}_\mathcal{D} P^{\mathsf{T}} \in \mathbb{R}^{d_{\texttt{proj}} \times d_{\texttt{proj}}}$ is an approximation to the Hessian in the projected space.

For the "default" values used for damping for TRAK, D-TRAK and K-FAC Influence, we primarily follow recommendations from prior work. For K-FAC Influence, the default is a small damping value $10^{-8}$ throughout added for numerical stability of inversion, as done in prior work (Bae et al., 2024). For TRAK-based methods, Park et al. (2023) recommend using no damping: "[...] computing TRAK does not require the use of additional regularization (beyond the one implicitly induced by our use of random projections)" (Park et al., 2023, § 6). Hence, we use the lowest numerically stable value of $10^{-9}$ as the default value throughout.

Note that all damping values reported in this paper are reported as if being added to the GGN for the Hessian of the loss *normalised by dataset size* . This differs from the damping factor in the TRAK implementation (https://github.com/MadryLab/trak), which is added to the GGN for the Hessian of an unnormalised loss ($\sum_n \ell(\theta, x_n)$). Hence, the damping values reported in (Zheng et al., 2024) are larger by a factor of $N$ (the dataset size) than the equivalent damping values reported in this paper.

### J.6 LDS BENCHMARKS

For all `CIFAR` LDS benchmarks Park et al. (2023), we sample 100 sub-sampled datasets ($M := 100$ in Equation (13)), and we train 5 models with different random seeds ($K := 5$ in Equation (13)), each with 50% of the examples in the full dataset, for a total of 500 retrained models for each benchmark. We compute the LDS scores for 200 samples generated by the model trained on the full dataset.

The only difference from the above for the `ArtBench` experiments is that we sample 50 sub-sampled datasets ($M := 50$ in Equation (13)). This gives a total of 250 retrained models for this benchmark.

**Monte Carlo sampling of measurements** For all computations of the "true" measurement functions for the retrained models in the LDS benchmarks we use 5000 samples to estimate the measurement.

### J.7 RETRAINING WITHOUT TOP INFLUENCES

For the retraining without top influences experiments (Figure 3), we pick 5 samples generated by the model trained on the full dataset, and, for each, train a model with a fixed percentage of most influential examples for that sample removed from the training dataset, using the same procedure as training on the full dataset (with the same number of training steps). We then report the change in the measurement on the sample for which top influences were removed.

**Monte Carlo sampling of measurements** Again, for all computations of the "true" measurement functions for the original and the retrained models used for calculating the difference in loss after retraining we use 5000 samples to estimate the measurement.

### J.8 TRAINING DETAILS

For `CIFAR-10` and `CIFAR-2` we again follow the training procedure outlined in Ho et al. (2020), with the only difference being a shortened number of training iterations. For `CIFAR-10`, we train for 160000 steps (compared to 800000 in Ho et al. (2020)) for the full model, and 80000 steps for the subsampled datasets (410 epochs in each case). On `CIFAR-2`, we train for 32000 steps for the model trained on the full dataset, and 16000 steps for the subsampled datasets ($\sim 800$ epochs in each case). We train for significantly longer than Zheng et al. (2024), as we noticed the models trained using their procedure were noticeably undertrained (some per-diffusion-timestep training losses $\ell_t(\theta, x)$ have not converged). We also use a cosine learning-rate schedule for the `CIFAR-2` models.

For `ArtBench-10`, we use the pretrained autoencoder from Stable Diffusion 2 (Rombach et al., 2022), but we train the diffusion backbone from scratch (initialise randomly). We follow the training procedure in (Rombach et al., 2022) and train the full model for 200000 training iterations, and the models trained on the subsampled data for 60000 iterations. We use linear warm-up for the learning rate schedule for the first 5% of the training steps. We use the `AdamW` optimiser with a learning rate of $10^{-4}$, weight-decay of $10^{-6}$, gradient norm clipping of 1, and exponential moving average (EMA) with maximum decay rate of 0.9999 and EMA warm-up exponential factor of 0.75 (see the https://github.com/huggingface/diffusers library for details on the EMA parameters). We don't use cosine learning rate decay. We only train the diffusion backbone, leaving the original pretrained autoencoder unchanged.

### J.9 HANDLING OF DATA AUGMENTATIONS

In the presentation in Section 2, we ignore for the sake of clear presentation the reality that in most diffusion modelling applications we also apply data augmentations to the data. For example, the training loss $\mathcal{L}_{\mathcal{D}}$ in Equation (3) in practice often takes the form:

$$\mathcal{L}_{\mathcal{D}} = \frac{1}{N} \sum_{n=1}^{N} \mathbb{E}_{\tilde{x}_n} \left[ \ell(\theta, \tilde{x}_n) \right],$$

where $\tilde{x}_n$ is the data point $x_n$ after applying a (random) data augmentation to it. This needs to be taken into account **1)** when defining the GGN, as the expectation over the data augmentations $\mathbb{E}_{\tilde{x}_n}$ can either be considered as part of the outer expectation $\mathbb{E}_z$, or as part of the loss $\rho$ (see Section 2.3), **2)** when computing the per-example train loss gradients for influence functions, **3)** when computing the loss measurement function.

When computing $\text{GGN}_{\mathcal{D}}^{\text{model}}$ in Equation (9), we treat data augmentations as being part of the out "empirical data distribution". In other words, we would simply replace the expectation $\mathbb{E}_{x_n}$ in the definition of the GGN with a nested expectation $\mathbb{E}_{x_n} \mathbb{E}_{\tilde{x}_n}$:

$$\text{GGN}_{\mathcal{D}}^{\text{model}}(\theta) = \mathbb{E}_{x_n} \left[ \mathbb{E}_{\tilde{x}_n} \left[ \mathbb{E}_{\tilde{t}} \left[ \mathbb{E}_{x^{(\tilde{t})}, \epsilon^{(\tilde{t})}} \left[ \nabla_\theta^\mathsf{T} \epsilon_\theta^{\tilde{t}} \left( x^{(\tilde{t})} \right) (2I) \nabla_\theta \epsilon_\theta^{\tilde{t}} \left( x^{(\tilde{t})} \right) \right] \right] \right] \right].$$

with $x^{(\tilde{t})}$ now being sampled from the diffusion process $q(x^{(\tilde{t})} | \tilde{x}_n)$ conditioned on the augmented sample $\tilde{x}_n$. The terms changing from the original equation are indicated in yellow. The "Fisher" expression amenable to MC sampling takes the form:

$$\text{F}_{\mathcal{D}}(\theta) = \mathbb{E}_{x_n} \left[ \mathbb{E}_{\tilde{x}_n} \left[ \mathbb{E}_{\tilde{t}} \left[ \mathbb{E}_{x_n^{(\tilde{t})}, \epsilon^{(\tilde{t})}} \mathbb{E}_{\epsilon_{\text{mod}}} \left[ g_n(\theta) g_n(\theta)^\mathsf{T} \right] \right] \right] \right], \qquad \epsilon_{\text{mod}} \sim \mathcal{N} \left( \epsilon_\theta^{\tilde{t}} \left( x_n^{(\tilde{t})} \right), I \right),$$

where, again, $g_n(\theta) = \nabla_\theta \|\epsilon_{\text{mod}} - \epsilon_\theta^{\tilde{t}}(x_n^{(\tilde{t})})\|^2$.

When computing $\text{GGN}_{\mathcal{D}}^{\texttt{loss}}$ in Equation (11), however, we treat the expectation over daea augmentations as being part of the loss $\rho$, in order to be more compatible with the implementations of TRAK (Park et al., 2023) in prior works that rely on an empirical GGN (Zheng et al., 2024; Georgiev et al., 2023).[16] Hence, the GGN in Equation (11) takes the form:

$$\text{GGN}_{\mathcal{D}}^{\texttt{loss}}(\theta) = \mathbb{E}_{x_n} \left[ \nabla_\theta \left( \mathbb{E}_{\tilde{x}_n} \left[ \ell(\theta, \tilde{x}_n) \right] \right) \nabla_\theta^\intercal \underbrace{\left( \mathbb{E}_{\tilde{x}_n} \left[ \ell(\theta, \tilde{x}_n) \right] \right)}_{\tilde{\ell}(\theta, x_n)} \right]$$

$$= \mathbb{E}_{x_n} \left[ \nabla_\theta \tilde{\ell}(\theta, \tilde{x}_n) \nabla_\theta^\intercal \tilde{\ell}(\theta, \tilde{x}_n) \right],$$

where $\tilde{\ell}$ is the per-example loss in expectation over data-augmentations. This is how the Hessian approximation is computed both when we're using K-FAC with $\text{GGN}_{\mathcal{D}}^{\texttt{model}}$ in presence of data augmentations, or when we're using random projections (TRAK and D-TRAK).

When computing the training loss gradient in influence function approximation in equation Equation (5), we again simply replace the per-example training loss $\ell(\theta^\star, x_j)$ with the per-example training loss averaged over data augmentations $\tilde{\ell}(\theta^\star, x_j)$, so that the training loss $\mathcal{L}_{\mathcal{D}}$ can still be written as a finite sum of per-example losses as required for the derivation of influence functions.

For the measurement function $m$ in Equation (6), we assume we are interested in the log probability of (or loss on) a particular query example in the particular variation in which it has appeared, so we do not take data augmentations into account in the measurement function.

Lastly, since computing the training loss gradients for the influence function approximation for diffusion models usually requires drawing MC samples anyways (e.g. averaging per-diffusion timestep losses over the diffusion times $\tilde{t}$ and noise samples $\epsilon^{(t)}$), we simply report the total number of MC samples per data point, where data augmentations, diffusion time $\tilde{t}$, etc. are all drawn independently for each sample.

---

[16] The implementations of these methods store the (randomly projected) per-example training loss gradients for each example before computing the Hessian approximation. Hence, unless data augmentation is considered to be part of the per-example training loss, the number of gradients to be stored would be increased by the number of data augmentation samples taken.

