# OpenReview forum: "Influence Functions for Scalable Data Attribution in Diffusion Models"
_ICLR.cc/2025/Conference — ICLR 2025 Oral_

### Official Review · Reviewer_VBdr · 2024-10-29

**Soundness:** 3
**Presentation:** 3
**Contribution:** 4
**Rating:** 10
**Confidence:** 4

**Summary:**

The paper provides a systematic treatment of tractable influence function (IF) estimation for generative diffusion models. Several forms of IFs are considered: IF of the ELBO, IF of the loss, IF of the path probabilities. These approaches are compared both conceptually and experimentally. The IF of the marginal probabilities of the images is discussed but not implemented.
A large fraction of the paper deals with the intractability of the Jacobian estimation involved in the calculation of the IF and offers and compares several approximate solutions.

The paper provides a thorough experimental analysis of the different IF approaches on several datasets and compares the results with the exact retraining method.

[edit] I increased my score due to the new analysis and results provided during the rebuttal phase, which answered some of the opened questions and provided links with the recent literature.

**Strengths:**

This work is very valuable for anyone interested in IF estimation in generative diffusion models. This is likely going to be a very important tool due to privacy and copyright concerns associated with image and video generation.

Both the theoretical discussion and the experimental analysis is systematic and well executed. The paper is very easy to follow and it manages to both provide the intuition and the theoretical justification for its design choices.

I particularly appreciated the discussion of the results, which highlight both the strengths and weaknesses of each approach and also the limitation of the approximate IF estimation when compared with exact retraining.

**Weaknesses:**

Arguably, the marginal likelihood logp(x) is the ideal target for a IFa analysis. While it is somewhat challenging to compute using the deterministic integration formula with stochastic or deterministic  trace estimation, it is not intractable and in my opinion it should have been analyzed. For example, having the analysis of the IF of logp(x) would help to make sense of the counterintuitive behavior of the IF of the ELBO.

It would have also appreciated to see an analysis on more tractable low-dimensional models with small networks, where the Hessian can be computed exactly. This analysis can be used to evaluate the relative performance of the Hessian approximation techniques, at least in this simple regime.

**Questions:**

In the last paragraph before the discussion (starting from line (507), the authors argue that the ELBO could be a poor proxy of the marginal probability due to the fact that it is rather insensitive to the resampling of up 50% of the dataset, meaning that such a resampling do not meaningfully change the ELBO of the generated images.

I think that this conclusion is incorrect. There is in fact evidence that diffusion training enters a 'generalization phase' after a finite dataset size, after which the generative probabilities remain approximately constant even under a non-overlapping split of the training set. This for example can be seen in (Zahra, 2023).

References:
Kadkhodaie, Zahra, et al. "Generalization in diffusion models arises from geometry-adaptive harmonic representation." arXiv preprint arXiv:2310.02557 (2023).

---

> ### Author Response · Authors · 2024-11-23
>
> Thank you for a detailed review and the feedback! We very much appreciate that you think the work is really valuable for anyone interested in influence functions in generative diffusion models! Here, we will try to address your outstanding points and describe what we have done, or are planning on doing, to address them.
>
> ### Weaknesses
>
> > Arguably, the marginal likelihood logp(x) is the ideal target for a IF analysis.
>
> Yes, you're right. Although deriving an influence function approximation for the $\log p(x)$ measurement would have been difficult (since computing $\nabla_\theta \log p(x)$ is challenging), we could have still considered using $\log p(x)$ as a ground-truth measurement.
>
> The primary reason we haven't done it is that, in the standard DDPM training setup, the data distribution doesn't have a density (the pixel values are discrete). This makes us worried about then interpreting the model trained on that data as a density model (the density values can be made arbitrarily high by just concentrating the density around the discrete pixel values everywhere). The DDPM ELBO acknowledges the fact that the data is discrete by including a reconstruction term (which, in all fairness, is also numerically quite poorly behaved). Hence, the DDPM ELBO targets a different “marginal log-probability” than that computed by the continuous normalising flow reformulation of the model (the latter is a marginal log-density). The works that compute the exact log-likelihoods in the diffusion settings usually train on uniformly dequantised data to get around this issue.
>
> We have CIFAR-2 models trained on uniformly dequantised data that we investigated in an unrelated ablation. We could look at how well influence functions/TRAK with ELBO/loss proxies approximate the exact log-likelihood in that setting?
>
> > It would have also [been] appreciated to see an analysis [...] with small networks, where the Hessian can be computed exactly.
>
> We're considering an analysis with a smaller network on CIFAR-2 ($<1$ million parameters), for which we can compute exact inverse Hessian vector products (and exact inverse GGN vector products) with iterative solvers. We're unsure if this will be computationally tractable for trained models that still learn something meaningful from the data, but we will report our results.
>
> ### Questions:
>
> > [...] the authors argue that the ELBO could be a poor proxy of the marginal probability [...] I think that this conclusion is incorrect. There is in fact evidence that diffusion training enters a 'generalization phase' after a finite dataset size, after which the generative probabilities remain approximately constant even under a non-overlapping split of the training set.
>
> We were not aware of the results from Zahra Kadkhodaie et al., thank you for pointing these out! The results in Kadkhodaie et al. seem to corroborate “Observation 3”, but we agree they would imply that the ELBO is not necessarily a bad proxy for marginal log-probability of generating a sample. We revised the discussion around “Observation 3” to reference the results from Kadkhodaie et al., and removed sentences conjecturing that observation 3 could imply that ELBO is a poor proxy.
>
> Furthermore, the results of computation of exact marginal probabilities on dequantised data might shed some extra light on this, since we'll be able to look at the correlation between the ground-truth ELBO and log-likelihood measurements (albeit in a slightly different setting).
>
> Given the results from Kadkhodaie et al., it seems likely that observation 3 will be dependent on subset size used for retraining. Therefore, we will also try to check what happens to the correlation of the ELBO measurements between retrained models as we vary the amount of training data. Currently, we are running retraining of models with random subsets of 10, 50, 500, 5000, and 25000 examples from CIFAR-10. These will hopefully allow us to make a closer connection between the generalisation results observed by Kadkhodaie et al. and our results with the ELBO. We will report these results and further revise Observation 3 discussion once they are available.

---

> > ### Comment · Reviewer_VBdr · 2024-11-26
> >
> > Thank you very much for the detailed reply. I do think that the new analysis would greatly strengthen the paper and I am looking forward to seeing the new results.
> > In particular, I do think that the analysis of log p(x) for de-quantized data will be very insightful and it will clarify the behavior of the ELBO.

---

> > > ### Author Response · Authors · 2024-12-01
> > > **New exact log p(x) experiments**
> > >
> > > > I do think that the analysis of log p(x) for de-quantized data will be very insightful and it will clarify the behavior of the ELBO.
> > >
> > > **We completed the LDS analysis on de-quantised CIFAR-2 with exact log-likelihood as the target ground-truth measurement.** The results are shown below (or you can see the figure [here](https://anonymous.4open.science/api/repo/iclr2025-rebuttal-supplementary-C798/file/lds_scores_log_likelihood.pdf))
> > >
> > > | Method | Rank Correlation % (LDS) | Untuned Result % |
> > > |--------|-------------------------|----------------|
> > > | CLIP Cosine Similarity | 4.1 ± 0.7 | n/a |
> > > | TRAK (m. ELBO) | 4.0 ± 0.7 | 1.0 |
> > > | TRAK (m. loss) | 10.4 ± 0.7 | 0.6 |
> > > | D-TRAK | 14.9 ± 0.8 | 0.5 |
> > > | K-FAC Influence (m. ELBO) | 9.6 ± 0.9 | 9.6 |
> > > | K-FAC Influence (m. loss) | 18.1 ± 0.8 | 18.1 |
> > > | Exact Retraining | 35.2 ± 0.4 | n/a |
> > >
> > > **We also investigated the behaviour of the ELBO, log-likelihood and the loss measurements as the amount of training data changes.** We wanted to see how these quantities relate as the model enters the generalisation phase (as seen in the reference you recommended – Kadkhodaie et al. (2024)).
> > >
> > > For this experiment, we retrained multiple models on different subsets of (de-quantised) CIFAR-10, varying the size of the training subset. We were interested in: 1) do the models exhibit a generalisation phase and a memorisation phase similar to Kadkhodaie et al. (2024), and 2) is this reflected in the ELBO and log-likelihood measurements on new samples?
> > >
> > > We do observe the same behaviour as Kadkhodaie et al. (2024) — the samples generated from the models are all different for models trained on subsets below a certain size, and they are nearly identical for models trained on subsets past a certain size (see [figure](https://anonymous.4open.science/api/repo/iclr2025-rebuttal-supplementary-C798/file/cifar10deq_log_likelihood_correlation_across_subset_sizes_with_samples.pdf)). This “collapse” also happens for the models' log likelihoods, as well as the ELBO. Past a certain subset size threshold, both the ELBO and the log-likelihoods of models retrained on different subsets become identical (see [this figure](https://anonymous.4open.science/api/repo/iclr2025-rebuttal-supplementary-C798/file/cifar10deq_simpelbo_correlation_across_subset_sizes.pdf) for the ELBO results). Before that subset size threshold, our “Observation 3” does not hold.
> > >
> > > That being said, the ELBO and the log-likelihood measurements are not identical, and ELBO is not a perfect proxy for the log-likelihood (as seen from the LDS results). We do observe that the ELBO starts becoming a better proxy for the log-likelihood once the training data sizes are sufficient for the model to enter the generalisation phase (see [figure](https://anonymous.4open.science/api/repo/iclr2025-rebuttal-supplementary-C798/file/cifar10deq_elbo_to_log_likelihood_correlation_across_subset_sizes.pdf)). The diffusion loss is sometimes a better proxy for the log-likelihood — in terms of correlation across non-training samples — for low training data sizes before the generalisation phase (see [figure](https://anonymous.4open.science/api/repo/iclr2025-rebuttal-supplementary-C798/file/cifar10deq_loss_to_log_likelihood_correlation_across_subset_sizes.pdf)).
> > >
> > > Hence, based on these results, our original speculation that ‘“Observation 3” casts into doubt the suitability of the ELBO as a proxy for marginal likelihood’ becomes uncompelling. On the other hand, these new results put into question the typically used design of the LDS evaluation of data attribution methods for diffusion models. If the subsets used for the LDS benchmarks are of sufficient size that the retrained models are in the generalisation phase, the differences in the measurements of the retrained models might be minute.

---

> > > > ### Comment · Reviewer_VBdr · 2024-12-02
> > > >
> > > > I wish to thank the authors for their work. I really appreciated the new analysis and results, which I think answer some of the key open questions in the original submission.
> > > > I will therefore increase my score. I recommend the authors to integrate these new results in the manuscript.

---

### Official Review · Reviewer_cA9c · 2024-11-01

**Soundness:** 3
**Presentation:** 3
**Contribution:** 3
**Rating:** 8
**Confidence:** 2

**Summary:**

In this work the authors adapt influence functions to work with diffusion models. Computing influence functions typically requires one to compute the inverse of the Hessian, which is intractable for deep learning models. Therefore the authors make use of a series of approximations. Firstly, they approximate the Hessian with the generalized Gauss-Newton matrix (GGN). Then they approximate the GGN with a block diagonal matrix, where the diagonal blocks are defined by layers. Finally, they approximate each block as a kronecker product.

The authors then evaluate their method on “linear data modelling score” and “retraining without top influencers”, and find their methods achieve state-of-the-art on both metrics.

**Strengths:**

This paper combines interesting ideas to tractably approximate influence functions for diffusion models, and achieve state-of-the-art performance on training data attribution on diffusion models, which is a challenging problem.

**Weaknesses:**

While it is clear that the proposed method is much more scalable than computing the raw influence function, it is difficult to assess how scalable this method actually is based on the main body of this paper (e.g. things like what is the runtime, how does it scale with the model size, what are the space requirements). A reader ought to have a good idea of how feasible this would be to implement for themselves.

It is not clear whether the proposed method outperforms the existing methods given the overlaps in standard error. Perhaps statistical significance could be reported here to give a better idea?

The experimental evaluation is only performed on CIFARs 2 and 10, so it is unclear how well this method scales to larger image generators, both in the quality of attributions and in terms of compute.

**Questions:**

You explicitly mention that K-FAC is defined for linear and convolutional layers (page 6), but this doesn’t seem to include attention layers, which all diffusion architectures that I’m aware of use. Is K-FAC also defined for attention in prior work, or could you define it?

Could you provide a runtime analysis, or some idea of what the runtime is (e.g. how much time would it take to compute a all influence values for a single sample on an A100)? How does this runtime (and possibly space requirement) scale with the complexity of the model? How much computational resources would I need if I wanted to run this on my favorite existing diffusion model?

---

> ### Author Response · Authors · 2024-11-23
>
> Thank you for a thorough review and the feedback! The overall verdict on the paper appears positive, which we very much appreciate. Here, we will try to address your outstanding points and describe what we have done, or are doing, to address them.
>
> ## Weaknessses
> > [...] it is difficult to assess how scalable this method actually is [...]. A reader ought to have a good idea of [what is the runtime, how does it scale with the model size, what are the space requirements].
>
> That's a good point. **We added an appendix section with the goal of giving the reader an impression of the runtime and memory requirements (new Appendix E)** of K-FAC Influence. We discuss both asymptotic compute and memory scaling of K-FAC Influence against TRAK, and report empirical runtimes for a single NVIDIA A100 for our experiments. The experiments for which we report the runtimes include ones on a more complex ArtBench dataset which we are currently running an LDS benchmark on.
>
> The new appendix section will, hopefully, also answer your question: *“Could you provide a runtime analysis, or some idea of what the runtime is [...]?”*.
>
> > It is not clear whether the proposed method outperforms the existing methods given the overlaps in standard error. Perhaps statistical significance could be reported here to give a better idea?
>
> The standard error bars are indeed overlapping in the “retraining without top influences” experiments in Figure 3. In response, we ran more experiments to double the number of retraining runs for that experiment. We updated Figure 3 in the paper.
>
> We find that the (one-sided) p-values for the paired difference test of the score across the 4 settings in Figure 3 being greater for K-FAC Influence are: 0.0003% when comparing against TRAK, and 1.7% when comparing against D-TRAK respectively. If you think it'd be informative, we can add the previous sentence to the experiments section.
>
> The p-values for the paired difference tests on individual benchmarks are:
> - CIFAR-2 — 2% removed | K-FAC Influence against TRAK: $p=0.00048$
> - CIFAR-2 — 10% removed | K-FAC Influence against TRAK: $p=0.0024$
> - CIFAR-10 — 2% removed | K-FAC Influence against TRAK: $p=0.093$
> - CIFAR-10 — 10% removed | K-FAC Influence against TRAK: $p=0.031$
> - CIFAR-2 — 2% removed | K-FAC Influence against DTRAK: $p=0.2$
> - CIFAR-2 — 10% removed | K-FAC Influence against DTRAK: $p=0.043$
> - CIFAR-10 — 2% removed | K-FAC Influence against DTRAK: $p=0.29$
> - CIFAR-10 — 10% removed | K-FAC Influence against DTRAK: $p=0.17$
>
> > The experimental evaluation is only performed on CIFARs 2 and 10
>
> You're right, it would be great to demonstrate our method on more complex datasets with larger models. To address this, we are currently running the LDS experiments with a larger latent diffusion model (LDM) on a higher resolution dataset. We are running the benchmarks on the full ArtBench dataset ($256\times 256$ resolution images). We will hopefully update the results in the paper within the discussion period.
>
> ## Questions
> > You explicitly mention that K-FAC is defined for linear and convolutional layers (page 6), but this doesn’t seem to include attention layers, which all diffusion architectures that I’m aware of use. Is K-FAC also defined for attention in prior work, or could you define it?
>
> Yes, K-FAC can be defined for all linear layers with weight sharing, including attention (Eschenhagen et al. (2023)). Example 1 in Section 2.2 of Eschenhagen et al. (2023) is a good explanation of how the attention mechanism can be thought of as a linear layer with weight sharing. Previously, in the main text body we said that we implement "K-FAC-expand whenever weight sharing is used" (line 292/293), but we have adjusted the text to explicitly mention attention.
>
> We have also added an additional appendix section which summarises the general K-FAC formulation for linear layers with weight sharing from previous work (new Appendix C).
>
>
> ---
>
> Thanks for the feedback. If you have any further thoughts on how to better incorporate the changes mentioned above, please let us know!

---

> > ### Comment · Reviewer_cA9c · 2024-11-26
> >
> > Thank you for your response, the p-values and the confirmation regarding attention layers is appreciated. Scaling up to ArtBench would in my opinion demonstrate that the method can feasibly scale to real datasets.

---

> > > ### Author Response · Authors · 2024-12-04
> > > **ArtBench Results**
> > >
> > > > Scaling up to ArtBench would in my opinion demonstrate that the method can feasibly scale to real datasets.
> > >
> > > We agree, and **we have successfully ran the LDS benchmark on the ArtBench dataset (`50000` images, `256x256` resolution) with unconditional Latent Diffusion Models (LDMs).** You can see the [updated results figure for the loss measurement here](https://anonymous.4open.science/api/repo/iclr2025-rebuttal-supplementary-C798/file/lds_scores_loss.pdf). We describe the results in more detail in a global response to all the reviewers.

---

### Official Review · Reviewer_nmGh · 2024-11-05

**Soundness:** 3
**Presentation:** 2
**Contribution:** 3
**Rating:** 6
**Confidence:** 4

**Summary:**

This paper extends influence functions for data attribution of diffusion models. With an exposition focusing on approximating a Hessian with a generalized Gauss-Newton matrix (GCN), this paper justifies specific design choices (linearizing the neural networks combined with K-FAC-expand approximation) that lead to better performing influence functions. The theoretical framework also subsumes previous work such as Journey-TRAK and D-TRAK. Empirically, influence functions perform similarly to D-TRAK on CIFAR-2 and CIFAR-10, with lower sensitivity to the choice of the damping parameter. Finally, empirical observations are made to show the challenges when applying influence functions to diffusion models.

**Strengths:**

- A theoretical framework is shown to unify influence functions, Journey-TRAK, and D-TRAK for diffusion models. This provides some clarity to the field of data attribution for diffusion models, which currently seems rather empirical.
- The theoretical framework motivates the design choices for approximating influence functions, which actually lead to better performance (as shown in Figure 4).
- Empirical observations (Section 4.1) are made to better understand how to apply influence functions to diffusion models.

**Weaknesses:**

- The LDS and counterfactual evaluations are limited to DDPM on CIFAR. The paper would be strengthened if K-FAC Influence is also evaluated on LDM and LoRA-fine-tuned Stable Diffusion (as in the Journey-TRAK and D-TRAK papers). Furthermore, since the observations in Section 4.1 are empirical, not theoretical, they need to be validated in other model architectures and datasets to make the claims more general.
- The proposed method K-FAC Influence is also sensitive to the choice of the damping parameter (LDS can range from ~15% to ~5% on CIFAR-10). This range of variation seems not much different from D-TRAK.
- Observation 3 might be sensitive to the amount of training data removed. For example, if 90% of the data are removed, there could be more variation in the retrained diffusion models, as suggested by Kadkhodaie et al. (2024; Figure 2 in particular). In this sense, the claim in Observation 3 might need to be adjusted to be less general.
- This is minor: It would be useful to include a brief summary of Grosse and Martens (e.g., the reduce vs. expand settings), perhaps in the Appendix.

References
 - Kadkhodaie et al. (2024) - Generalization in Diffusion Models Arises from Geometry-Adaptive Harmonic Representations.

**Questions:**

- In practice, how would we tune the damping parameter for K-FAC Influence? How would the tuning procedure differ from that for D-TRAK?
- Observation 2 shows that F-FAC Influence overestimates the impact of data removal, whereas Koh et al. (2019) show that influence functions underestimates the impact of data removal. What are possible explanations for these differing observations?
- Observation 3 seems to have been shown in Kadkhodaie et al. (2024; Figure 2 in particular). What additional insights are provided? For example, does Observation 3 only hold for ELBO, or does it also hold for the simple diffusion loss?


References
- Koh et al. (2019) - On the Accuracy of Influence Functions for Measuring Group Effects.
- Kadkhodaie et al. (2024) - Generalization in Diffusion Models Arises from Geometry-Adaptive Harmonic Representations.

---

> ### Author Response · Authors · 2024-11-15
>
> Thank you for taking the time to write a thorough review and for all the suggestions! In the following, we will address each listed weakness, describe what we will aim to do about it, and answer the questions that were raised.
>
> ## Weaknesses
>
> > The LDS and counterfactual evaluations are limited to DDPM on CIFAR.
>
> You're right, we agree that our conclusions could be strengthened by considering more complex datasets and latent diffusion models (LDM).
>
> We will do the LDS experiments with an LDM model on a higher resolution dataset, and, hopefully, update the results within the discussion period. We are running the benchmarks on the full ArtBench dataset with a larger, unconditional LDM model trained from scratch. We were thinking of first focusing on unconditional LDMs to limit the number of changes to the previous settings.
>
> > The proposed method K-FAC Influence is also sensitive to the choice of the damping parameter [...].
>
> That is a fair point. However, while both methods might benefit from tuning, K-FAC Influence seems to work more consistently well with a default value. Choosing the smallest possible damping value that is numerically stable appears to be a good heuristic, and the value of `1e-8` used in prior non-diffusion literature works well, so it appears like a good universal default (see Figure 7 & 9). For TRAK these two points do not hold (see Figure 8 & 10). Lastly, TRAK-based methods do not work at all if the damping is too small, and what “too small” means seems to vary between the datasets considered.
>
> We decided to tone down the claim that “TRAK and D-TRAK appear to be significantly more sensitive to tuning the damping factor” (line 374) by dropping “significantly”. We would also welcome other suggestions for how to appropriately adjust the wording to describe these differences. Lastly, we updated Figure 2 to make the comparison of what one can expect with tuned and untuned damping factors a little bit clearer.
>
> > Observation 3 might be sensitive to the amount of training data removed [...] as suggested by Kadkhodaie et al. (2024) [...]
>
> We were not aware of the results from Kadkhodaie et al. (2024), thank you for pointing these out! These seem to corroborate “Observation 3”, but we agree they would imply that it might be dependent on the subset size. We revised the discussion around “Observation 3” to reference the results from Kadkhodaie et al., and adjusted the wording to make it clear the observation might depend on the dataset size.
>
> We will also try to check what happens to the correlation of the ELBO measurements between retrained models as we vary the amount of training data. Currently, we are running retraining of models with random subsets of 10, 50, 500, 5000, and 25000 examples from CIFAR-10. We will report these results and further revise Observation 3 discussion once they are available.
>
> > It would be useful to include a brief summary of Grosse and Martens (e.g., the reduce vs. expand settings) [...].
>
> We will add a more detailed explanation of K-FAC for the general case of linear layers with weight sharing to the appendix before the end of the discussion phase.

---

> > ### Author Response · Authors · 2024-11-15
> >
> > ## Questions
> >
> > > In practice, how would we tune the damping parameter for K-FAC Influence? How would the tuning procedure differ from that for D-TRAK?
> >
> > As far as we are aware, D-TRAK doesn't come with a recommended procedure for selecting the damping factor in a practical setting. One could try to follow what they do in the paper, i.e. select the damping factor which gives the highest LDS score on a dataset of interest. However, computing the LDS score requires tens or hundreds of training runs, so this is likely intractable for most settings of interest. The same holds for K-FAC Influence.
> >
> > Another option would be to use values that have worked on similar problems in the past. As stated above, this is where D-TRAK and K-FAC Influence seem to differ: the damping value for K-FAC Influence used in previous non-diffusion settings (`1e-8`) is also close to optimal across different datasets in our ablations (see Figure 7 & 9). In previous works on TRAK-based methods (other than D-TRAK) no damping has been used, but in the diffusion setting we see using low damping values results in weak performance for TRAK (Figure 8 & 10). As mentioned above, to better illustrate this, we adjusted Figure 2 to also show the LDS score with a small default damping value, as recommended in prior literature, in addition to the best performing one.
> >
> > > Observation 2 shows that F-FAC[K-FAC] Influence overestimates the impact of data removal, whereas Koh et al. (2019) show that influence functions underestimates the impact of data removal. What are possible explanations for these differing observations?
> >
> > Thanks for pointing this out, this is an interesting observation. We now cite Koh et al. (2019) and added this observation to the discussion in the appendix. We are uncertain about the exact reason for this discrepancy. There are several differences between the two works for (approximately) computing inverse Hessian products that could contribute to the observed behaviour:
> >
> > Firstly, Koh et al. (2019) (if we understand correctly) use iterative solvers to compute exact inverse Hessian-vector products, whereas we use an approximation of the GGN.
> > While the GGN is guaranteed to be positive semi-definite, the Hessian might have negative eigenvalues. For self-influence, making all the eigenvalues positive would strictly increase the estimated influence. However, this is a heuristic argument, so it would be interesting to compute and compare influence functions with exact inverse Hessian-vector products and inverse GGN-vector products for a smaller model.
> >
> > Secondly, another potential factor could be how Koh et al. and us handle weight decay and its influence on damping. Koh et al. consider a regularised loss function with a weight decay factor $\lambda$ (Equation (1)), which will be equivalent to the damping factor used for the Hessian (see $H_{\lambda, 1}$ under Equation (4)). In contrast, we use AdamW to train the model (as is standard for DDPMs), and can therefore not straightforwardly interpret weight decay to be part of the loss function. Consequently, the weight decay factor we use will not necessarily correspond to the damping factor used for our Hessian approximation.
> >
> > There are also many other differences between the two settings that might lead to a discrepancy: we use a layer-wise block-diagonal K-FAC approximation; we consider diffusion models (for which the stable score behaviour described by Kadkhodaie et al. (2024) could play a role in why actual changes in measurement are so small); and likely many more.

---

> > > ### Author Response · Authors · 2024-11-15
> > >
> > > > Observation 3 seems to have been shown in Kadkhodaie et al. (2024; Figure 2 in particular). What additional insights are provided? For example, does Observation 3 only hold for ELBO, or does it also hold for the simple diffusion loss?
> > >
> > > Our observations do appear very similar to those of Kadkhodaie et al. (2024), although they look at generated samples, whereas we look at the behaviour of the ELBO and the diffusion loss. We will make sure to reference and discuss Kadkhodaie et al. (2024) appropriately.
> > >
> > > Interestingly, our observation does, to an extent, also hold for the simple diffusion loss when we look at random subsets of data. We added a plot (Figure 18) to the paper equivalent to Figure 17, but for the simple diffusion loss. For CIFAR-10, the correlation is again around 100% between the retrained models, but for CIFAR-2 it is significantly lower. This seems to match the implication of Kadkhodaie's work that trained diffusion models start to behave the same for sufficiently large training set sizes. It's interesting that the threshold is different for the diffusion loss and the ELBO. This might imply that the correlations in per-diffusion-timestep losses transition to 100% at different dataset sizes, depending on the diffusion timestep. Similarly, Kadkhodaie et al. (Figure 1) implies that models transition from memorization to generalization for different dataset sizes at different diffusion timesteps, but this is not the same as all models trained on different subsets having the same loss at a particular diffusion timestep.
> > >
> > > Are you interested in any particular questions about this? We would be happy to explore this further.

---

> ### Author Response · Authors · 2024-12-01
> **New results on “Observation 3” and comparisons to Kadkhodaie et al. (2024)**
>
> > Observation 3 might be sensitive to the amount of training data removed.
>
> > Observation 3 seems to have been shown in Kadkhodaie et al. (2024). [...]  does Observation 3 only hold for ELBO, or does it also hold for the simple diffusion loss?
>
> Given the results by Kadkhodaie et al. (2024), we further investigated “Observation 3” and the behaviour of the ELBO as the training data size varies. **Whereas  Kadkhodaie et al. (2024) looked at the “collapse” to identical generated samples across different retrained models, we added new experiments that look at the exact log-likelihood, ELBO, and diffusion loss measurements most relevant to the influence setting.**
> Indeed, we do observe that “Observation 3” is subset size dependent, and holds for other measurements like exact log-likelihood or diffusion loss (although the transition occurs marginally ‘earlier’ for the ELBO than for the diffusion loss).
>
> For these experiments, we retrained multiple models on subsets of varying size of (de-quantised) CIFAR-10.
>
> We do reproduce the same behaviour as Kadkhodaie et al. (2024) — the samples generated from the models are all different for models trained on subsets below a certain size, and they are nearly identical for models trained on subsets past a critical size (see [figure](https://anonymous.4open.science/api/repo/iclr2025-rebuttal-supplementary-C798/file/cifar10deq_log_likelihood_correlation_across_subset_sizes_with_samples.pdf)). This “collapse” also happens for the models' log likelihoods, as well as the ELBO and the diffusion loss. Past a certain subset size threshold, both the ELBO, the loss and the log-likelihood of models retrained on different subsets become identical (see the [ELBO results figure](https://anonymous.4open.science/api/repo/iclr2025-rebuttal-supplementary-C798/file/cifar10deq_simpelbo_correlation_across_subset_sizes.pdf), and the [loss results figure](https://anonymous.4open.science/api/repo/iclr2025-rebuttal-supplementary-C798/file/cifar10deq_loss_correlation_across_subset_sizes.pdf)). Before that subset size threshold, our “Observation 3” does not hold. The threshold seems to be smaller for the ELBO compared to the loss measurement, and therefore we didn't see perfect correlation among the loss measurements on in our original experiments.
>
> That being said, the ELBO and the log-likelihood measurements are not identical, and ELBO is not a perfect proxy for the log-likelihood (as seen from the LDS results). We do observe that the ELBO starts becoming a better proxy for the log-likelihood once the training data sizes are sufficient for the model to enter the generalisation phase (see [figure](https://anonymous.4open.science/api/repo/iclr2025-rebuttal-supplementary-C798/file/cifar10deq_elbo_to_log_likelihood_correlation_across_subset_sizes.pdf)).
>
> ---
> We will revise section 4.1 to include a discussion of these new results. In particular, we have revised “Observation 3” to say:
> > For sufficiently large training set sizes, ELBO, log-likelihood and the diffusion loss are close to constant on generated samples, irrespective of which examples were removed from the training data.
>
> We will also start off the discussion of “Observation 3” by pointing out Kadkhodaie et al. (2024) were the first to notice this kind of “collapse”. We merely verify that it holds for the ELBO, log-likelihood and the diffusion loss, which has implications for using and interpreting influence functions in this setting.

---

> > ### Comment · Reviewer_nmGh · 2024-12-03
> >
> > The authors' responses have addressed my concerns about Weaknesses 2-4 and Questions 1-3.
> >
> > The result that influence functions can have reasonable LDS performance with the default damping parameter $\lambda = 1e^{-8}$ is particularly striking--because TRAK-based methods for diffusion models indeed require a surprisingly large damping parameter. In this sense, influence functions require less tuning and is more intuitive to use in practice. I encourage the authors to highlight this point in their paper (even simply as a footnote if space is a constraint).
> >
> > Overall, I raised my rating from a 5 to 6. The score is not higher because currently all the empirical results are about the CIFAR datasets. For example, it still remains possible that, on a different dataset and model architecture, the damping parameter needs to be set to a large value as for D-TRAK.

---

> ### Author Response · Authors · 2024-12-04
> **ArtBench**
>
> > Overall, I raised my rating from a 5 to 6.
>
> Thank you for reading and considering our response carefully. We're glad that the response has addressed many of your concerns!
>
> > The score is not higher because currently all the empirical results are about the CIFAR datasets. [...] it still remains possible that, on a different dataset and model architecture, the damping parameter needs to be set to a large value as for D-TRAK.
>
> We agree, and we've been working on new experiments to address this.
> **We successfully ran the LDS benchmark in a more challenging setting on the ArtBench dataset (`50000` images, `256x256` resolution) with unconditional Latent Diffusion Models (LDMs).** You can see the [new results figure for the loss measurement here](https://anonymous.4open.science/api/repo/iclr2025-rebuttal-supplementary-C798/file/lds_scores_loss.pdf). We describe the results in a global response to all the reviewers. In short: K-FAC matches TRAK with a matching measurement function, and, after introducing eigenvalue correction for K-FAC recommended in prior work, we observe that eigenvalue-corrected K-FAC improves upon TRAK/D-TRAK in all settings.
>
> In this new ArtBench LDM setting with a new dataset and architecture, we observe similar sensitivity to the damping factor selection: K-FAC/EK-FAC influence functions can have good LDS performance with the default damping parameter $\lambda=1e^{-8}$. You can see the ArtBench damping ablation for [TRAK and D-TRAK in this figure](https://anonymous.4open.science/api/repo/iclr2025-rebuttal-supplementary-C798/file/artbench_lds_trak_ablation.pdf) and for [K-FAC Influence in this figure](https://anonymous.4open.science/api/repo/iclr2025-rebuttal-supplementary-C798/file/artbench_lds_influence_ablation.pdf). We agree this is a strong argument for the use of K-FAC Influence in practice, and we will highlight this point in the paper as you recommended.
>
>
> Overall, we would like to again thank the reviewer for the feedback and comments on the paper, and suggestions for this and other experiments that we believe have made the paper more complete.

---

### Author Response · Authors · 2024-12-04
**ArtBench Results**

As requested by the reviewers, **we ran the LDS scoring in a more challenging setting on the ArtBench dataset (`50000` images, `256x256` resolution) with unconditional Latent Diffusion Models (LDMs).** You can see the [new results figure for the loss measurement here](https://anonymous.4open.science/api/repo/iclr2025-rebuttal-supplementary-C798/file/lds_scores_loss.pdf). We also summarise these ArtBench LDS results below:

| Group | Method | Rank Correlation % (LDS) | Untuned Result |
|-------|--------|-----------------|----------------|
|  | CLIP Similarity | 5.2% ±1.0 | - |
| Correct measurement | TRAK | 9.8% ±1.0 | 7.5% |
| ↳ | **K-FAC Influence** | 10.0% ±0.9 | 8.7% |
| ↳ | **EK-FAC Influence** (eigenvalue-corrected K-FAC) | 16.9% ±0.8 | 15.5% |
| Incorrect measurement | D-TRAK | 13.8% ±1.0 | 1.0% |
| Baseline | Exact Retraining | 21.3% ±0.3 | - |

Firstly, we have updated our K-FAC results with the eigenvalue correction from George et al. (2018) [1], as recommended for influence functions in Grosse, Bae and Anil et al. (2023) [2].  Eigenvalue-corrected Kronecker-factored Approximate Curvature (EK-FAC) is a minor modification to the K-FAC method that adjusts the eigenvalues of the approximation to closer match the approximated GGN. In all experiments, we adjust the number of samples used for K-FAC computation to hold the computational cost approximately fixed (half as many samples for EK-FAC as for K-FAC). The eigenvalue correction appears to consistently and significantly improve the quality of the influence approximation in all our experiments.

Secondly, we found a bug in a public library on which we have been relying for our experiments, which adversely affected the previously reported K-FAC LDS scores. The bug led to the effective damping for bias parameters being higher than for weight parameters. With this fix, we get slightly improved performance on all LDS benchmarks (see the [updated results figure](https://anonymous.4open.science/api/repo/iclr2025-rebuttal-supplementary-C798/file/lds_scores_loss.pdf)). Morevoer, we observe even higher robustness to the choice of the damping factor for EK-FAC Influence (see [this CIFAR-2 figure](https://anonymous.4open.science/api/repo/iclr2025-rebuttal-supplementary-C798/file/cifar2_lds_damping_ablation_bias_fix.pdf) and [this CIFAR-10 figure](https://anonymous.4open.science/api/repo/iclr2025-rebuttal-supplementary-C798/file/cifar10_lds_damping_ablation_bias_fix.pdf)). The previously shared log-likelihood LDS benchmark figure already includes this fix.

We have integrated these new results into the paper and have expanded the summary of prior work on K-FAC in Appendix C to cover EK-FAC. We will also update the new log-likelihood LDS results figure with the ArtBench experiments once those are finished.

[1] George et al. (2018), "Fast Approximate Natural Gradient Descent in a Kronecker-factored Eigenbasis"

[2] Grosse, Bae and Anil et al. (2023), “Studying Large Language Model Generalization with Influence Functions”


---


Lastly, we'd like to thank all the reviewers for the feedback on the paper. Interesting points that we haven't considered before have been raised in the reviews, and the paper has become significantly more well-rounded thanks to the discussion and experiments we have included thanks to the reviewers' suggestions.

---

### Meta-Review · Area_Chair_ndQP · 2024-12-20

**Metareview:**

This paper extends influence functions to diffusion models, addressing the computational challenges of Hessian inversions with scalable approximations, including Kronecker-factored Approximate Curvature (K-FAC) and its eigenvalue-corrected variant (EK-FAC). The proposed method demonstrates state-of-the-art performance on benchmarks like Linear Data Modeling Score (LDS). The authors provide theoretical insights that unify influence functions with prior work (e.g., TRAK and D-TRAK).

**Additional Comments On Reviewer Discussion:**

Reviewers noted the paper’s strong theoretical foundation, scalability, and contributions to understanding influence functions in diffusion models. Initial concerns about limited dataset coverage, computational scalability, and sensitivity to hyperparameters were addressed through experiments, including on larger datasets like ArtBench. The authors' responses also clarified the runtime requirements and resolved discrepancies in evaluation metrics. Notably, the inclusion of EK-FAC improved results across settings. All reviewers supported acceptance, with some recommending additional clarifications in the final version.

---

### Decision · Program_Chairs · 2025-01-22

Accept (Oral)